# Random Label Prediction Heads for Studying Memorization in Deep Neural Networks

**Marlon Becker**    **Jonas Konrad**    **Luis Garcia Rodriguez**    **Benjamin Risse**
University of Münster, Germany
{marlonbecker,jonas.konrad,luis.garcia,b.risse}@uni-muenster.de

## Abstract

We introduce a straightforward yet effective method to empirically study memorization in deep neural networks for classification tasks. Our approach augments each training sample with auxiliary random labels, which are then predicted by a random label prediction head (RLP-head). RLP-heads can be attached at arbitrary depths of a network, predicting random labels from the corresponding intermediate representation and thereby enabling analysis of how memorization capacity evolves across layers. By interpreting the RLP-head performance as an empirical estimate of Rademacher complexity, we obtain a direct measure of both sample-level memorization and model capacity. We leverage this random label accuracy metric to analyze generalization and overfitting in different models and datasets. Building on this approach, we further propose a novel regularization technique based on the output of the RLP-head, which demonstrably reduces memorization. Interestingly, our experiments reveal that reducing memorization can either improve or impair generalization, depending on the dataset and training setup. These findings challenge the traditional assumption that overfitting is equivalent to memorization and suggest new hypotheses to reconcile these seemingly contradictory results. The source code is available at https://github.com/MarlonBecker/RandomLabelHeads.

## 1 Introduction

Modern deep learning models are prone to overfitting due to their extreme over-parameterization (Nakkiran et al., 2021). A wide range of strategies have been proposed to mitigate this issue, including data augmentation, explicit regularization, and dataset scaling. Although enlarging training datasets has proven particularly effective, this approach is often infeasible in domains where data acquisition or annotation is expensive or requires significant human expertise. Moreover, existing strategies primarily address practical concerns of generalization but provide limited insight into the mechanisms by which overfitting arises.

Recent work highlights the striking memorization capacity of state-of-the-art models. For instance, Zhang et al. (2021) demonstrate that modern architectures can perfectly fit datasets with randomly assigned labels, thereby achieving 100 % training accuracy in the absence of any learnable structure. In such cases, high accuracy is attainable only through memorization of individual training samples, underscoring that contemporary artificial neural networks (ANNs) can encode sample-specific and task-irrelevant information to fit each training sample individually.

This ability to memorize arbitrary labels is directly connected to the model complexity. In particular, training with SGD on random labels empirically approximates Rademacher complexity, which plays a central role in deriving generalization bounds within the PAC-learning framework.

The primary objective of this work is to assess the accuracy of predicting random labels as a practical metric of memorization. Although direct training on random labels reveals a model's ability to memorize, this procedure does not intrinsically inform how memorization interacts with generalization in real-world tasks and does not allow memorization mitigation. To bridge this gap, we propose a hybrid approach: we augment the network with an additional Random Label Prediction Head (RLP-head), attached to the feature extractor (i.e., all layers except the final classification layer) in parallel to the original task head, which remains unchanged. This design enables simultaneous measurement and

regularization of memorization during normal training, thereby providing a controlled way to study and modulate memorization in deep neural networks. In summary, our contribution is as follows:

- We propose the use of random label prediction heads (RLP-heads) as a tool for probing layer-wise memorization in deep neural networks.
- We validate that the random label accuracy derived from RLP-heads is an accurate measure for complexity and memorization.
- We propose a novel regularizer that explicitly constrains memorization by penalizing the performance of the RLP-head during training.
- Building on our metric and regularizer, we show how memorization can hinder or, in certain scenarios, facilitate generalization. We further hypothesize that this dual role is driven by sampling effects in the training data.

## 2 RELATED WORK

The phenomenon of data memorization, although not new, gained renewed attention in the era of modern deep learning with the works of Zhang et al. (2021) and Arpit et al. (2017). Traditionally, memorization was associated with model capacity and overfitting, and hence viewed primarily as a source of poor generalization. This view of capacity being responsible for overfitting has been challenged by the discovery of the double descent phenomenon (Nakkiran et al., 2021), which reveals a more nuanced relationship between capacity and generalization.

Feldman (2019) formalize memorization as the ability of a model to correctly predict a label only if the sample was present in the training data. Their analysis suggests that the key obstacle to generalization is not label noise but suboptimal sampling, with many regions of the data distribution undersampled or represented by only a single example. We compare our proposed memorization metric in detail to the work of Feldman & Zhang (2020) in Appendix A.12. Even though these atypical examples in so-called long-tailed data distributions are memorized individually to reach high training performance, this memorization leads to improved generalization of the network (cf. Feldman & Zhang (2020)).

Building on this perspective, Baldock et al. (2021) observe that deep models first capture simple patterns shared across many examples, before gradually fitting more complex patterns that may be unique to a small subset of the data or even example-specific. A similar observation can be found in Liu et al. (2020), where the authors develop a framework to leverage that property to be able to learn in noisy scenarios. Bayat et al. (2024) argue that memorization is not inherently detrimental, but rather depends on factors such as data quality and learning dynamics. They introduce the notion of an example-specific feature rate, showing that excessively high rates prevent models from capturing the underlying distribution, while excessively low rates encourage the learning of overly complex representations, leading to catastrophic overfitting.

Subsequent work examined memorization, including studies by Carlini et al. (2019) and Yun et al. (2019), with particular attention to the effects of heavy overparameterization (Zhang et al., 2020) and minimal overparameterization (Daniely, 2020). Another line of research examines where memorization occurs within a network. For instance, Maini et al. (2023) demonstrate that memorization is localized across layers and even within specific neurons. Our approach is closely aligned with this perspective: by attaching RLP-heads at different layers, we obtain a direct means of localizing memorization.

Memorization effects are particularly pronounced in large-scale language models, where they raise significant privacy concerns if training data can be extracted from the models, as highlighted by Tirumala et al. (2022) and Carlini et al. (2021). Efforts to improve generalization and mitigate data memorization have largely focused on general-purpose regularization methods, such as dropout (Srivastava et al., 2014) and weight decay (Krogh & Hertz, 1991). However, to the best of our knowledge, no existing approach explicitly regularizes memorization itself, as we propose in this work.

Closely related challenges arise in the context of fair AI, where suppression of unwanted or spurious features is critical to prevent models from encoding biases related to attributes such as gender, ethnicity, or religion (Mehrabi et al., 2021; Tian et al., 2022; Wang et al., 2020; Zhang et al., 2018a). We take technical inspiration from this field to develop our memorization suppressing regularizer. Finally, our interpretation of random label accuracy as a proxy for information abstraction bears conceptual

resemblance to mutual information frameworks, which have been applied to analyze ANNs (Gabrié et al., 2018).

## 3  BACKGROUND: RADEMACHER COMPLEXITY

We take inspiration from the Rademacher complexity measure to motivate our empirical metric. Rademacher complexity is a fundamental tool in statistical learning theory, quantifying the expressive power of a model (or hypothesis class) by measuring its ability to fit random labels. In the case of binary classification, it can be defined as follows:

**(Empirical) Rademacher complexity for Binary Classification (Mohri et al., 2012):** Given a hypothesis class $\mathcal{H}$ and train data $\mathcal{S} = \{(x_1, \sigma_1), ..., (x_m, \sigma_m)\}$, where $\sigma_1, ..., \sigma_m \in \{\pm 1\}$ are i.i.d. uniform random variables:

$$\hat{\mathfrak{R}}_{\mathcal{S}}(\mathcal{H}) = \mathbb{E}_{\sigma}\left[\sup_{h \in \mathcal{H}} \frac{1}{m} \sum_{i=1}^{m} \sigma_i h(x_i)\right] \tag{1}$$

In binary classification, the agreement between a model's prediction and the true label can be quantified by the product of the label and the model output. While this measure is closely related to accuracy, it is inherently restricted to the binary setting and does not naturally extend to multi-class classification. The hypothesis $h$ is chosen as a supremum over the hypothesis class, which in practice can be approximated via empirical risk minimization (e.g., with optimizers such as SGD or Adam). However, the presence of the supremum makes the exact evaluation of Rademacher complexity intractable in practical settings.

Importantly, it is model-agnostic, and therefore explicitly independent of architectural details including depth, width, and the total number of parameters. Instead, it captures the capacity of a model through its ability to fit random labels. Within the PAC-learning framework, this quantity is central to deriving bounds on the generalization error. In particular, for binary classification, the generalization error can be bounded as:

*Theorem* 1. Given a hypothesis class $\mathcal{H}$, training data $\mathcal{S} = \{(x_1, \sigma_1), ..., (x_m, \sigma_m)\}$, with $\sigma_1, ..., \sigma_m \in \{\pm 1\}$, then for any $\delta > 0$, with probability at least $1 - \delta$ for any $h \in \mathcal{H}$ it holds that

$$R(h) \leq \hat{R}_{\mathcal{S}}(h) + \hat{\mathfrak{R}}_{\mathcal{S}}(\mathcal{H}) + 3\sqrt{\frac{\log(2/\delta)}{2m}}.$$

Where $\hat{R}_{\mathcal{S}}(h)$ denotes the empirical error on the training dataset (Mohri et al., 2012). This bound implies that, for fixed training performance, a reduction in Rademacher complexity directly translates into improved test performance bounds and thus tightens limits on the generalization error. While Rademacher complexity provides a theoretically powerful framework for characterizing the capacity of hypothesis classes, its exact computation for state-of-the-art deep learning models is infeasible.

Inspired by this theoretical foundation, we will derive an empirical alternative to Rademacher complexity, suited for real-world training tasks, thereby enabling the study of the relation between memorization and generalization in practical deep learning settings.

## 4  RANDOM LABEL PREDICTIONS AND REGULARIZATION

Rather than training an entire network on random labels, as explored in prior work, we introduce an auxiliary Random Label Prediction Head (RLP-head) that predicts a randomly assigned label in parallel with the standard classification task. Concretely, the proposed architecture outputs both the task prediction vector $p \in \mathbb{R}^N$ and an additional random label prediction vector $\hat{p} \in \mathbb{R}^n$. While the number of task classes $N$ is determined by the dataset, the number of possible random labels $n$ can be chosen arbitrarily. The RLP-head may be attached at different locations within the network. Unless otherwise specified, we place it after the penultimate layer, in parallel with the standard classification head. This choice is natural since the penultimate activations correspond to the final stage of the feature extractor, and the RLP-head thereby probes the extent of memorization within the learned final representation.

Random labels are generated once at the beginning of the training and remain fixed across epochs for each sample. Only the RLP-head receives gradients from the random label objective, ensuring that the normal classification head is unaffected. Consequently, our method enables probing memorization

without affecting normal task performance.

In order to train the RLP-head we introduce an auxiliary cross-entropy loss on the random labels, $L^{rnd}$, in addition to the standard classification loss, $L^{class}$, where $y$ denotes the correct class label and $\hat{y}$ the assigned random label:

$$L^{class} = -\sum_{i=1}^{N} \delta_{iy} \log(p_i) = -\log(p_y) \quad (2)$$

$$L^{rnd} = -\sum_{i=1}^{n} \delta_{i\hat{y}} \log(\hat{p}_i) = -\log(\hat{p}_{\hat{y}}) \quad (3)$$

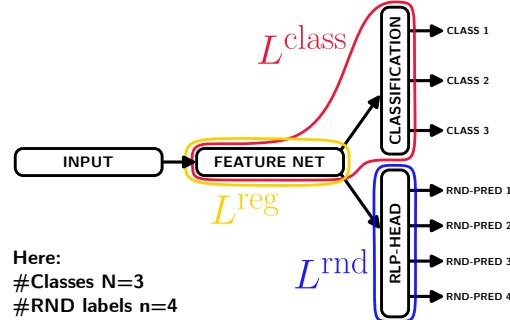

Here:
#Classes N=3
#RND labels n=4

By default, we implement the RLP-head as a single fully-connected layer followed by a softmax activation. Nevertheless, the architecture of the RLP-head is flexible, and more complex variants can be used (see Appendix A.6 for results with a two-layer head).

Figure 1: An additional Random Label Prediction Head (RLP-head) is added after the feature extractor of the network. Only the RLP-head receives $L^{rnd}$, the random label prediction loss, whereas the regularizing loss $L^{reg}$ is calculated on the RLP-head but acts on the feature extractor only.

Training the RLP-head on random labels in parallel with the main task enables to directly regularize memorization during standard training. Since we interpret the accuracy of the random label prediction head as an empirical proxy of the Rademacher complexity, regularizing the random label predictions provides a means of constraining the effective complexity of the model.

Therefore, we introduce a regularization loss term that penalizes correct predictions of the random labels by the RLP-head. Specifically, this loss is derived from the standard cross-entropy formulation:

$$L^{reg} = \sum_{i=1}^{n} \delta_{i\hat{y}} \log(1 - \hat{p}_i) = \log(1 - \hat{p}_{\hat{y}}). \quad (4)$$

Compared with standard cross-entropy, we invert the sign of the loss, since the regularizer is designed to prevent the network from learning the random labels. Furthermore, we replace $\hat{p}_i$ with $1 - \hat{p}_i$ inside the logarithm, which amplifies the penalty when $\hat{p}_i \approx 1$. This ensures that highly confident predictions of random labels are penalized more strongly. The resulting regularization term is scaled by a tunable hyperparameter $\lambda$ and added to the loss of the feature extractor.

Although the regularization loss is computed using the RLP-head, its gradients are restricted to the feature extractor. Accordingly, the classification head remains unaffected during RLP-regularization. A schematic of the proposed architecture is provided in Figure 1. Conceptually, the RLP-head and the feature extractor form two adversarial components: the RLP-head attempts to fit the random labels, while the feature extractor is regularized to prevent this from happening. This adversarial setup encourages the feature extractor to produce representations that are less example-specific and do not allow memorization of specific inputs. The proposed regularizer is, therefore, used here as a tool to investigate the effects of memorization in different parts of the network.

## 5 EXPERIMENTS

Details of our experimental setup can be found in Appendix A.1.

### 5.1 LEARNING RANDOM LABELS

Throughout this section, we analyze the training of the RLP-head, such that it serves solely as a metric and does not influence network performance (i.e., $\lambda = 0$). Figure 2A shows the test and train accuracy of the classification head alongside the random label accuracy extracted from the RLP-head for ViT-B/32 trained on ImageNet. Around epoch 20, test and train accuracies begin to diverge, indicating the beginning of overfitting. Notably, the random label accuracy starts to rise slightly earlier, reaching approximately 70 % by the end of training. This shows that, even when trained exclusively on correct class labels, the model memorizes a substantial portion of the dataset enough

for a single fully-connected layer to correctly predict random labels. The fact that random label accuracy does not approach 100 % may reflect that the chosen network architecture does not have sufficient capacity to fully memorize the dataset, consistent with the training accuracy plateauing at roughly 93 %.

Since we train the RLP-head together with the main classifier, we cannot tell whether its low early-epoch accuracy is due to the RLP-head not having been trained long enough or because the network has not yet memorized many samples. To disentangle these effects, we performed an

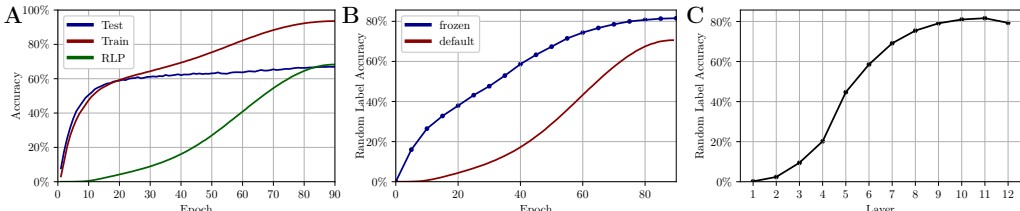

Figure 2: ViT-B/32 on ImageNet. **A**: The proposed single fully-connected layer as RLP-head is sufficient to correctly predict approx. 70 % of the random labels, indicating that the feature extractor memorizes a substantial portion of the training set. **B**: Even after freezing the feature extractor, RLP-head attains low accuracy in the early epochs, confirming that the default RLP-head approach reliably tracks the evolution of memorization dynamics during training. **C**: Random label accuracy when attaching the RLP-head at various network depths. The higher accuracy observed in deeper layers indicates that increasingly abstract representations still retain sample-specific information allowing for memorization.

additional experiment with two different modes of training the RLP-head shown in Figure 2B. *Default* refers to training the RLP-head in parallel with the main task, as described previously. *Frozen* refers to freezing all layers except the RLP-head at checkpoints saved after each epoch, and subsequently training (only) the RLP-head *from scratch*. For all frozen runs, the RLP-head is initialized with the same parameters and trained on the same fixed set of random labels. This setup ensures that the RLP-head receives sufficient and equal training capacity at each epoch. At epoch 0 (random weights), the frozen training fails to fit the random labels, indicating that the signal measured by the default training actually stems from memorization learned by the feature extractor during training and not from limitations of the RLP-head. Although this frozen training does not allow for regularization and is computationally very demanding, it is shown here to validate the suitability of our proposed default training method.

We further investigate where memorization occurs within the network by attaching a separate RLP-head consisting of a normalization layer, a fully-connected layer and a softmax layer after each transformer block of a ViT-B/32 trained on ImageNet (Figure 2C). The random label accuracy increases with the network depth: After the first layer nearly 0 % of the random labels can be predicted correctly, while high accuracies are reached in later layers. Similar to the previous experiment, this dependency shows that the RLP-head does not itself memorize the input sample but instead reflects the representational properties of the network. After the first layer, where only minimal processing occurred, the activations retain a significant amount of sample-specific information. Interestingly, this does not lead to an increased random label accuracy. Instead, high random label accuracies are reached only after sufficient abstraction of the features, showing that the abstracted features are still sample-specific and lead to memorization.

## 5.2 Relation to Other Complexity Regularizers

We propose to use the accuracy of the RLP-head as a proxy for model complexity, providing an empirical approximation to Rademacher complexity. To validate this interpretation, we evaluate our metric under three well-established regularization strategies, namely dropout, weight decay, and label smoothing. As illustrated in Figure 3, each of these regularizers consistently suppresses random label accuracy, confirming the correlation of the random label accuracy with model complexity.

We further support this correlation by studying the impact of the model size on the random label accuracy in Appendix A.9. We also use the random label accuracy to demonstrate that mixup reduces - but does not fully eliminate - memorization in Appendix A.16. Additional experiments with ViT-S/32

on ImageNet comparing the proposed random label accuracy against measuring memorization via noisy labels for different regularizers can be found in Appendix A.14.

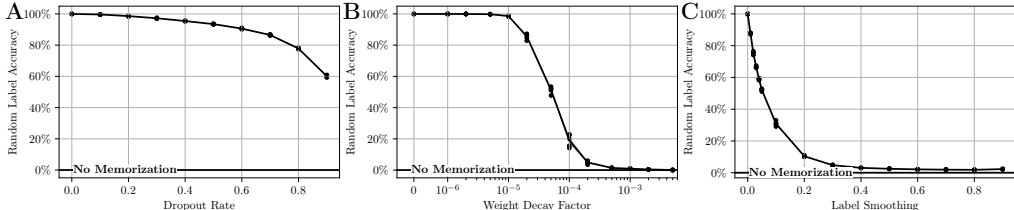

Figure 3: WRN16-4 on CIFAR-100. The effect of common complexity regularizers can be measured with the proposed metric. **A:** Dropout. **B:** Weight decay. **C:** Label smoothing.

## 5.3 REGULARIZING RANDOM LABELS

We can use the RLP-head to explicitly regularize the memorization of the network. To accomplish this, we apply the loss term defined in Equation 4 and search for an optimal regularization factor $\lambda$. We report results for ViT-B/32 on ImageNet in Figure 4 and WideResNet-16-4 on CIFAR-100 in Figure 5. We find that RLP-regularization effectively suppresses memorization in both experimental

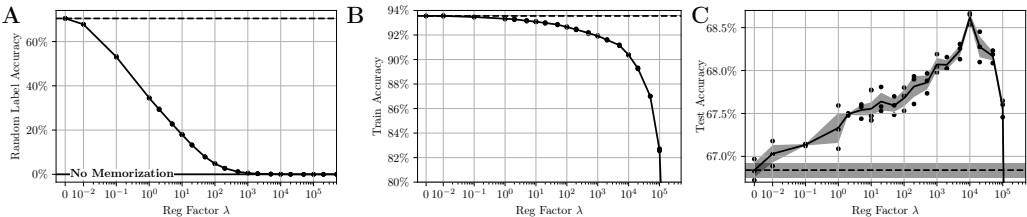

Figure 4: ViT-B/32 on ImageNet. Random label, train and test accuracy under RLP-regularization for different regularization factors $\lambda$. RLP-regularization effectively reduces memorization, and leads to better generalization (smaller test-train gap) and test performance.

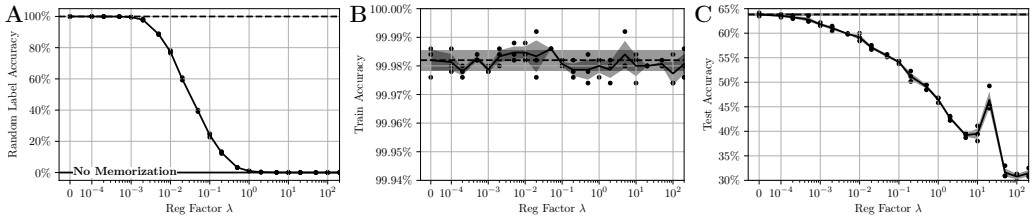

Figure 5: WideResNet-16-4 on CIFAR-100. Random label, train and test accuracy under RLP-regularization for different regularization factors $\lambda$. Here, RLP-regularization effectively reduces memorization, but does not improve generalization.

settings reducing the random label accuracy down to the level expected from random guessing. On ImageNet with ViT, this effect translates into improved generalization: while training accuracy decreases, test accuracy increases, reaching a peak of 68.5 % at $\lambda = 10^4$, which corresponds to a gain of 1.5 % over the baseline. The simultaneous drop in training accuracy further narrows the train–test gap, confirming the effectiveness of RLP-regularization to reduce overfitting. These observations align with predictions from PAC-learning theory based on Rademacher complexity, as well as the intuition that memorization causes overfitting and harms generalization.

Interestingly, these findings do not hold for our experiments for WideResNet-16-4 on CIFAR-100. Instead, the training accuracy remains unaffected, while the test accuracy deteriorates even for small regularization factors. These deviations from classical theory are consistent with recent findings, e.g., by (Nakkiran et al., 2021), which highlight the distinct dynamics of modern overparameterized

networks. Our results suggest that the relationship between memorization and generalization is more nuanced than traditional theory predicts, which we study further in the following sections.

## 5.4 Undersampled Datasets Benefit from Memorization

Based on our findings and drawing on insights from Feldman (2019) and Bayat et al. (2024), we hypothesize two distinct memorization scenarios that reconcile the apparent contradictions with the classical view of overfitting.

Memorization corresponds to the adoption of features that are highly specific to individual samples. Suppressing memorization prevents the learning of sample-specific features, forcing it instead to focus on features shared across examples of the same class. When sufficient samples are available, this results in learning features of the underlying true data distribution leading to increased generalization (cf. Figure 6A). Without memorization, training accuracy decreases because the network may fail to fit atypical samples, especially those that share few features with other samples in the same class, such as noisy or mislabeled samples. We hypothesize that this mechanism explains the observed behavior on ImageNet (Figure 4). However, when the dataset is undersampled and memorization is suppressed,

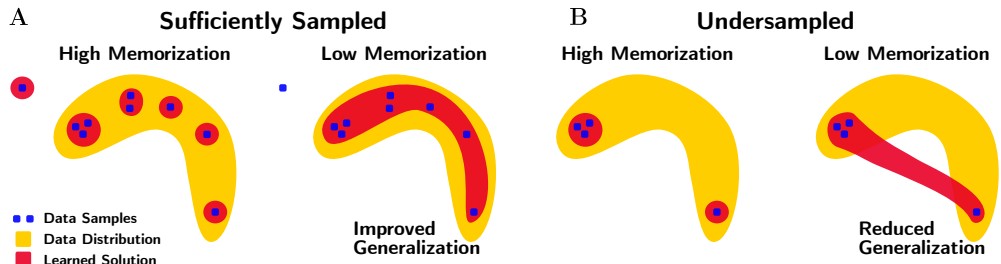

Figure 6: Schematic illustration of how memorization can be either detrimental or benign depending on dataset sampling. Under memorization, the model learns sample-specific solutions (depicted as small isolated regions around individual samples). In contrast, suppressing memorization encourages the discovery of a single connected solution space that better captures class-level structure while excluding outliers such as noisy or mislabeled samples.

the shared features learned across class samples may fail to reflect the true data distribution and instead capture arbitrary artifacts of the insufficient sampling. In this case, suppressing memorization forces the network to rely on these spurious shared features, which degrades generalization. New, unseen samples may still resemble individual memorized training examples but are unlikely to share the learned spurious features shared by training examples from undersampled regions of the true data distribution (cf. Figure 6B). We hypothesize that this mechanism explains the behavior observed on CIFAR-100 (Figure 5). In line with this view, we find the same effect (reduced random label accuracy, stable training accuracy, and degraded test accuracy) when applying the RLP-regularizer to ViT trained on CIFAR-100 (Appendix A.3).

To further test this hypothesis, we study the impact of dataset size by training ViT-B/32 on subsets of ImageNet while keeping the experimental setup fixed. As shown in Figure 7, our regularizer improves test accuracy only when large fractions of the dataset are available. The conventional intuition that memorization is always detrimental would suggest that reducing memorization should be even more beneficial on smaller datasets, where higher memorization (as observed by higher random label accuracy) occurs. Our experiment thus provides evidence in support of our hypothesis of beneficial memorization effects for undersampled datasets.

We perform an additional experiment where we inject label noise into the training dataset and apply the RLP-regularizer. Since noisy labels cannot contribute positively to generalization and can only be fit through memorization, our regularizer should consistently improve test performance in this setting. This prediction is confirmed in Figure 7C.

Related findings were also reported by Feldman (2019), who argue that memorization in sparsely sampled regions of the data distribution (i.e., the long tail) can actually enhance generalization. Because the proposed RLP-regularizer directly suppresses memorization, we apply it to the ImageNet-LT dataset (Liu et al., 2019b) to demonstrate in Appendix A.19 that classes in the long tail (i.e., those with few training samples) can no longer be predicted correctly when memorization is inhibited.

Taken together, our experiments highlight both detrimental and beneficial aspects of memorization

and demonstrate that RLP-heads, along with the derived regularizer, provide an effective framework for probing and controlling these dynamics.

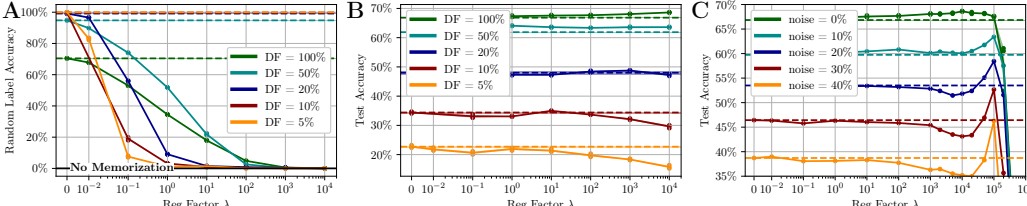

Figure 7: ViT-B/32 on ImageNet. **A+B**: Random label and test accuracy when training on reduced dataset fractions (DF) of ImageNet. Although smaller training sets lead to stronger memorization (higher random label accuracy), suppressing memorization on them does not improve test accuracy. **C**: Test accuracy with added label noise under RLP-regularization. Since memorization of noisy labels hinders generalization, our regularizer yields substantial improvements.

## 5.5 RLP-REGULARIZATION SHIFTS MEMORIZATION

To further understand the effects of the RLP-regularizer, we analyze memorization across different layers of the network. We attach additional RLP-heads after each layer of a vision transformer, as described above. Figure 8A shows the resulting random label accuracy across layers for varying regularization strengths.

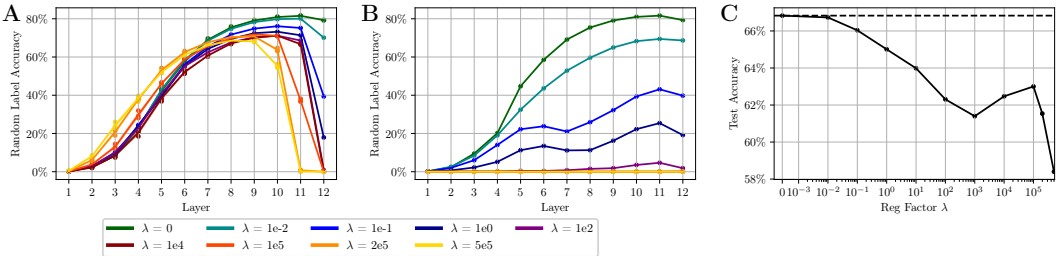

Figure 8: ViT-B/32 on ImageNet. **A**: Random label accuracy of RLP-heads at different layers when only the final (12th) layer is used for RLP-regularization. Memorization shifts toward earlier layers. **B+C**: RLP-regularization is calculated based on RLP-heads attached to all 12 transformer layers. While this effectively suppresses memorization and prevents the shift, neither test accuracy nor generalization improve.

The RLP-regularization is only applied based on the RLP-head attached to the final (12th) layer. Consequently, the random label accuracy drops rapidly for this last layer with increasing regularization. RLP-heads near the regularized final layer, particularly layers 10 and 11, are also affected. In contrast, earlier layers exhibit the opposite effect: RLP-heads attached to layers 2 to 6 achieve higher random label accuracies under regularization. This indicates that while memorization is mitigated in the last layer, it is shifted to earlier layers rather than eliminated. We hypothesize that, in response to the RLP-regularizer, the network transforms sample-specific features into class-relevant information in earlier layers, thereby enabling memorization to persist while being undetected by the regularizing RLP-head attached to the final layer.

To test this hypothesis, we conduct an additional experiment, adding a classification head to each transformer layer trained to predict the class label. This setup enables tracking the transformation from sample-specific features to class information throughout the network. Figure 9 shows the resulting class, train, and test accuracies under RLP-regularization based on the final layer. While class accuracy decreases in the last layer and the penultimate layer (11), we observe increased accuracy in earlier layers for both training and test data. Remarkably, test accuracies at layers 10 and 11 even surpass those of layer 12 (Figure 9C), indicating that regularization not only shifts memorization and classification capabilities but can also improve generalization in earlier layers.

This supports the hypothesis that RLP-regularization shifts the transformation into class-specific information to earlier layers.

Next, we examine the effect of suppressing memorization when using all attached RLP-heads for our regularization. As shown in Figure 8B, this effectively reduces random label accuracy at all layers, even for modest regularization strengths. However, this does not translate into improved test accuracy (Figure 8C). We hypothesize that applying RLP-regularization to all layers constitutes an overly harsh intervention: Extraction of sample-specific features in early layers may be useful even when these features do not lead to direct memorization. Moreover, some degree of memorization may persist within a transformer block itself, being hidden to the respective RLP-head attached at its end. We further study this hypothesis in Appendix A.20.

Additionally, we study the influence of the regularizer when the loss term is constructed from a single RLP-head attached to an intermediate layer in Appendix A.10.

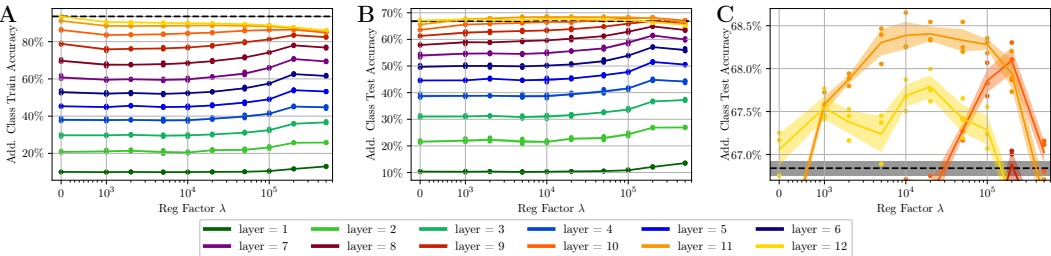

Figure 9: ViT-B/32 on ImageNet. Similar to the RLP-heads, we attach additional classification heads to the outputs of all layers in a ViT to track the transformation from sample-specific features to class predictions throughout the network. When applying RLP-regularization to the final (12th) layer only, class prediction accuracy increases in the earlier layers and test performance improves across all layers. **A**: Train accuracy. **B**: Test accuracy. **C**: Zoomed-in view of test accuracy.

## 6 CONCLUSION

We have introduced an effective method to measure and regularize memorization in deep neural networks: random layer prediction heads (RLP-heads), which can be attached to any (intermediate) network activation. Motivated as an empirical approximation of Rademacher complexity, we demonstrated that random label accuracy serves as a valid metric for network complexity and memorization. This metric enables the study of both the temporal (i.e., during optimization) and spatial (i.e., across layers) dynamics of memorization within a network. Based on the RLP-heads, we derived a regularization method to explicitly mitigate learning of sample-specific features and in consequence stop memorization.

Our experiments show that memorization can be either beneficial or detrimental for generalization deep neural networks. We propose a hypothesis to explain this counterintuitive effect based on dataset sampling and support it with targeted experiments. Moreover, applying the memorization regularizer to the final layer shifts both abstraction of class-level representations and memorization into earlier layers, resulting in a network that achieves better generalization after fewer layers.

Our findings highlight the value of RLP-heads and RLP-regularization for studying memorization and suggest their broader potential for empirical analysis of deep learning mechanisms.

## ACKNOWLEDGEMENTS

This work was funded by the German Research Foundation (DFG CRC 1459 Intelligent Matter - Project-ID 433682494).

Calculations for this publication were performed on the HPC cluster PALMA II of the University of Münster, subsidised by the DFG (INST 211/667-1).

## REPRODUCIBILITY STATEMENT

For all experiments, we report complete results, including the outcomes of all hyperparameter searches. Details on training configurations are provided in Appendix A.1. The source code is available at `https://github.com/MarlonBecker/RandomLabelHeads`.

## LLM USAGE

Large language models (LLMs) were used exclusively to assist in refining the phrasing of certain sentences and improving the clarity of formulations in this manuscript. At no point were LLMs employed for data analysis, generation of scientific content, or drawing conclusions. All scientific claims, results, and interpretations are the sole work of the authors.

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

APPENDIX

### A.1 EXPERIMENTAL SETUP

In the main text we focus on two evaluation scenarios:

1. WideResNet-16-4 (Zagoruyko & Komodakis, 2016) on CIFAR-100 (Krizhevsky & Hinton, 2009) trained with SGD with momentum $\mu = 0.9$, a linear learning rate warm up in the first epoch followed by a cosine decay with base learning rate of $\eta = 0.5$ and a batch size of 256 trained for 200 epochs without additional regularization or data augmentation. We use $n = 10,000$ as the number of (different) random labels.

2. ViT-B/32 (Dosovitskiy et al., 2021) on ImageNet-1k (Deng et al., 2009) trained with AdamW (Loshchilov & Hutter, 2019) and learning rate warm up for eight epochs followed by a cosine decay with base learning rate of $\eta = 0.001$ and a batch size of 1024 trained for 90 epochs with flipping augmentation, gradient clipping ($\ell_{max}^2 = 1.0$) and weight decay of 0.1. We use $n = 100,000$ random labels.

A full implementation comprising all models and configuration files is available at `https://github.com/MarlonBecker/RandomLabelHeads`.

### A.2 SANITY CHECK: SHUFFLED RANDOM LABELS

To validate that the observed increase of generalization actually stems from the mitigated memorization and is not a mere artifact, e.g., caused by effects on the scale of the feature activations, we perform a simple sanity check. We reshuffle all random labels in each epoch. Thus, the random labels cannot be learned and cannot serve as a metric for memorization. Consequently, the RLP-regularizer also does not explicitly reduce the memorization, while all other implicit effects of the regularizer remain. Results compared to our initially proposed RLP-regularization are shown in Figure A.1. As expected, the random label accuracy remains approximately at the chance of random guessing $1/n$. The train accuracy exhibits only a minor drop for high regularization factors. The test accuracy does not improve and is only affected by high regularization factors where the performance drops. This validates that the observed regularization is an effect of the mitigated memorization.

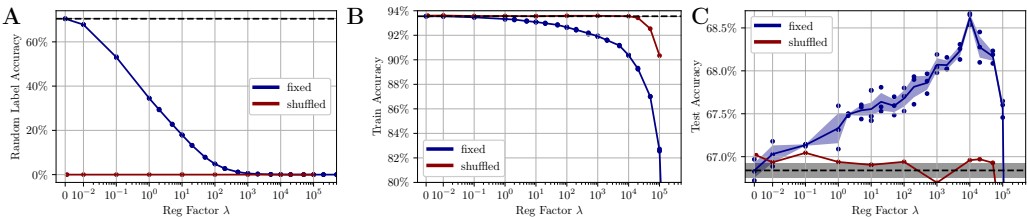

Figure A.1: ViT-B/32 on ImageNet. As a sanity check we compare the regularization results for fixed random labels (as before) to random labels shuffled in each epoch.

### A.3 VIT ON CIFAR-100

In section 5.3, we found opposing effects caused by memorization mitigation for our two experiments performed with ViT on ImageNet and with WideResNet on CIFAR-100. To clarify if the two observed effects are caused by the different model architectures or datasets, we perform an additional experiment where we study a ViT-S/4 trained on CIFAR-100. Results are shown in Figure A.2. Memorization is effectively stopped for regularization factors $\lambda > 10^{-1}$. Similarly to our experiments with WideResNet on CIFAR-100, we observe a detrimental effect of reducing memorization (unaffected training accuracy and reduced test accuracy) indicating the dataset to be pivotal for the different effects of memorization as we further examine in section 5.4.

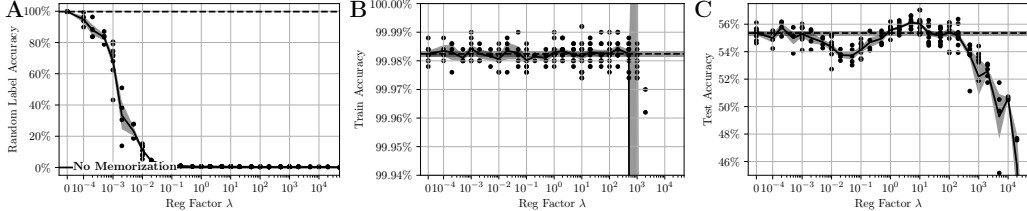

Figure A.2: ViT-S/4 on CIFAR-100 with RLP-regularization.

## A.4 NUMBER OF RANDOM LABELS

In Figure A.3 we analyze the effect of the number of different random labels $n$ when using a linear RLP-head. The input to the random prediction head, i.e., the feature dimension, stays constant and since the output of the linear layer is given by the number of random labels $n$, the capacity of the prediction head is directly tied to the number of random labels. Two intuitive implications can be directly observed from Figure A.3: The probability to reach high values by chance decreases with increasing $n$, i.e., the task to predict the random labels gets harder, and the capacity of the RLP-head grows with increasing $n$, i.e., the capability of the RLP-head to solve the given task increases. As a result, the reached random label accuracy undergoes a minimum before it approaches full memorization and saturates. From this experiment, we conclude that the number of random labels must be sufficiently large to ensure that the RLP-head has enough capacity to measure the models memorization. However, increasing the number of random labels $n$ also substantially raises computational costs. Balancing these considerations, we set $n = 10{,}000$ for WideResNet experiments on CIFAR-100 and $n = 100{,}000$ for ViT experiments on ImageNet.

To further validate our design choice on ImageNet, we additionally study the case where each training sample is assigned a unique random label (i.e., $n = m = 1{,}281{,}167$), and analyze the resulting effect of RLP-regularization in Appendix A.5.

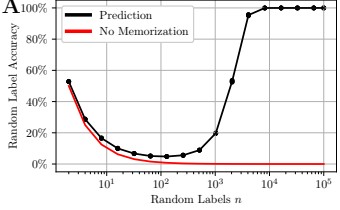 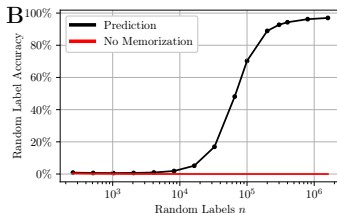

Figure A.3: Linear RLP-head. A sufficiently large number of random labels $n$ and thus head size has to be chosen. **A**: WRN16-4 on CIFAR-100. **B**: ViT-B/32 on ImageNet. The minimum is barely observable because the data starts at $n = 256$.

## A.5 UNIQUE LABEL PER SAMPLE

We compare our proposed random label formulation with the alternative of assigning a unique label to each sample. While the latter is computationally very expensive, it provides a direct measure of single-sample memorization. As shown in Figure A.4, unique labels yield higher memorization accuracy, due to the increased predictive capacity of the linear RLP-head. However, we observe no qualitative differences compared to our proposed approach with $n = 100{,}000$ labels. For computational efficiency, we adopt the latter approach in our experiments.

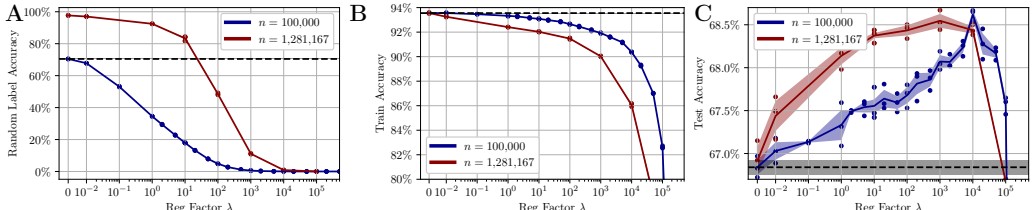

Figure A.4: ViT-B/32 on ImageNet. Using a unique label per sample when applying RLP-regularization (i.e., $n = m = 1{,}281{,}167$) compared to $n = 100{,}000$ used in the main paper.

## A.6 TWO-LAYER HEAD

As shown in the last section (Appendix A.4) the RLP-head capacity and the number of random labels $n$ are tied together for a linear head. To disentangle these two effects, we extend the RLP-head by adding a hidden fully-connected layer. We keep the number of labels constant at a rather small value in the experiment depicted in Figure A.5 ($n = 10$), and only influence the RLP-head capacity by varying its hidden feature dimension $d_h$. As can be seen, the RLP-head is now able to recover a much higher amount of random labels from the output of the corresponding feature extractor for a sufficiently large $d_h$, compared to the setting without a hidden layer in the RLP-head.

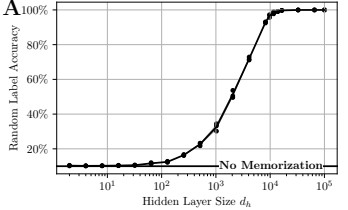
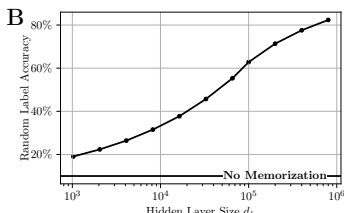

Figure A.5: A: WideResNet-16-4 on CIFAR-100. B: ViT-B/32 on ImageNet. RLP-head with one hidden layer. Number of random labels $n = 10$. Increasing the capacity of RLP-head leads to correctly predicted random labels.

Additionally, we do a sensitivity analysis on both the regularization strength controlled by $\lambda$ and the hidden layer size $d_h$ for WideResNet-16-4 on CIFAR-100. As shown in Figure A.6C small hidden layer dimensions have less impact on the test accuracy; however, the RLP-head is not capable to correctly predict the random labels under these conditions (see Figure A.6A). Larger RLP-heads do predict the random labels correctly and are thus sufficiently powerful to measure the network's memorization, but are similarly detrimental to the models generalization. Adding a hidden layer to the RLP-head used for regularization neither improves generalization nor yields qualitatively new insights. We therefore use a linear RLP-head.

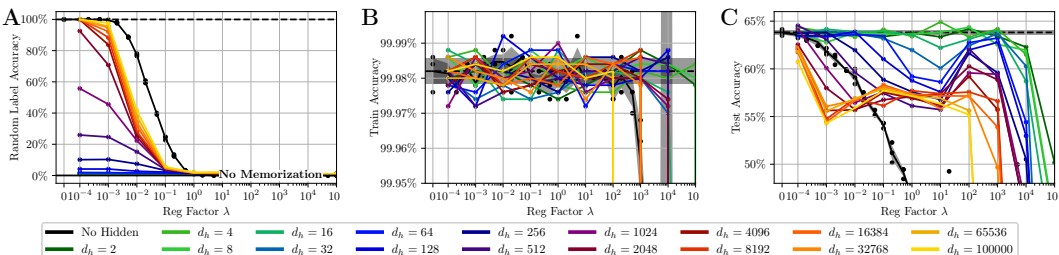

Figure A.6: WideResNet-16-4 on CIFAR-100. RLP-head with one hidden layer, $n = 100$.

## A.7 DATASET SIZE

Having studied the influence of the RLP-head in previous sections (Appendix A.4 and Appendix A.6), we aim to study the influence of the dataset size while maintaining the number of random labels and the capacity of the RLP-head constant now. We thereby ablate the influence of the dataset size on the difficulty of random label prediction task. We randomly sample subsets from CIFAR-100 in order to construct several smaller datasets and use a small RLP-head with $n = 1024$. As shown in A.7, the RLP-head is only able to predict the random labels correctly for small dataset sizes. Since we showed before that a large RLP-head can reach 100 % random label accuracy on the full dataset, the reduced random label accuracy is caused by the limited size of the RLP-head. We conclude from this experiment, that the needed RLP-head size to obtain adequately measure the memorization in the network grows with the dataset size.

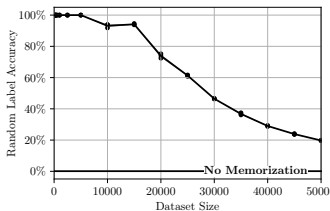

Figure A.7: WideResNet-16-4 on CIFAR-100. Dependence of random label accuracy on the dataset size for a small linear RLP-head of size $n = 1024$. The original dataset size of 50,000 training samples of CIFAR-100 is reduced by sampling random subsets.

## A.8 MULTI-HEAD RLP

While we aim to measure single-sample memorization, we chose to generate a number of $n$ random labels for $m$ total training samples, i.e., $m/n$ samples per random label, where usually $n \ll m$. For instance, we chose $n = 100,000$ for the $m = 1,281,167$ samples of ImageNet leading to approx. 12 images which attain the same random label. This results in the RLP-regularizer to only be able to effectively suppress features which are shared by parts of these random subsets of input images. While setting $n = m$ (that is, learning an individual random label per sample) is studied in Appendix A.5, this is not computationally feasible in practical scenarios. However, in the setting $n \ll m$, it is harder for the RLP-head to identify sample-specific features (as opposed to those shared in the random groups of images with the same random labels). This might allow the network to memorize sample-specific features even though the RLP-regularizer is applied. To circumvent this problem, we add multiple parallel RLP-heads receiving different sets of random labels. The total regularization loss is the average of the individual regularization losses per RLP-head. This way, a Multi-Head-RLP is developed which we hypothesize to be more powerful in identifying the networks memorization. However, it is computationally more demanding. As shown in Figure A.8, the number of heads in a multi-head setting does not impact the random label or train accuracy, but interestingly, yields even higher test accuracy, reaching 69.2 % for 10 heads and $\lambda = 10^4$.

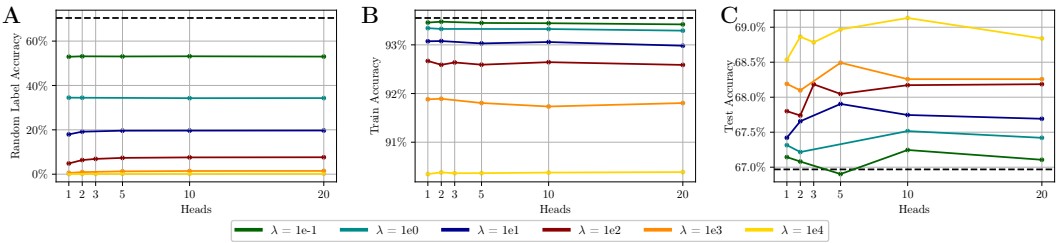

Figure A.8: ViT-B/32 on ImageNet. RLP-head used for regularization comprised of multiple parallel linear layers, each receiving a different mapping from random labels to input images. The multi-head structure results in improved generalization.

### A.9 FEATURE EXTRACTOR SIZE

To validate the proposed random label accuracy as a capacity metric, we analyze the impact of the feature extractor size on this measure. Specifically, we report the random label accuracy when training WideResNet-16-$w$ models with varying widening factors $w$ on CIFAR-100, without applying RLP-regularization (see Figure A.9). For small values of $w$, the models exhibit insufficient capacity to fully memorize the training data, which is directly reflected in lower random label accuracy. As $w$ increases, the models progressively achieve higher random label accuracy, until reaching a plateau at 100 %, indicating complete memorization of the dataset. These results support the use of random label accuracy, as measured by the RLP-head, as a reliable indicator of model capacity and complexity.

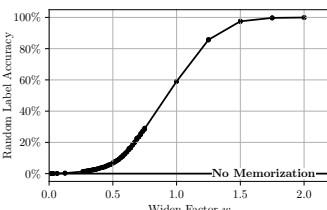

Figure A.9: WideResNet-16-$w$ on CIFAR-100, $n = 10,000$.

### A.10 REGULARIZING INTERMEDIATE LAYERS

Our proposed RLP-regularizer enables control over memorization in a layer-selective manner. To demonstrate this, we attach RLP-heads to all layers of a ViT trained on ImageNet (as in the main paper's section 5.5) to be able to monitor memorization across all layers, while we exclusively use the RLP-head at layer 8 for regularizing the full feature extractor. As can be seen in Figure A.10, the effect of the regularizer is highly localized: the random label accuracy at layer 8 is strongly suppressed, approaching zero under large regularization strengths. In contrast, adjacent layers exhibit only minor reductions in random label accuracy, and quickly recover beyond the regularized layer. Interestingly, despite employing a single intermediate layer for regularization, the test accuracy improves to a degree comparable to using RLP-regularization with the final layer, indicating enhanced generalization.

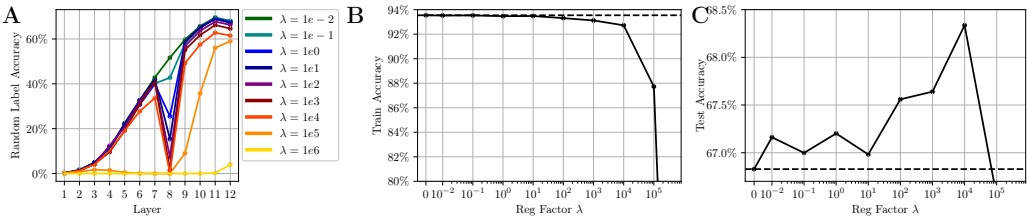

Figure A.10: ViT-B/32 on ImageNet. Only layer 8 is used for regularization.

### A.11 TEST-TRAIN DUPLICATES

Barz & Denzler (2020) show that CIFAR-100 contains numerous duplicates between the training and test sets. This phenomenon provides a plausible explanation for the negative effect of the RLP-regularizer's memorization reduction on test performance: duplicated test samples implicitly reward memorization of the training set. To address this issue, Barz & Denzler (2020) introduce a de-duplicated variant, ciFAIR-100, in which all duplicated test images are replaced by newly sampled datapoints.
We evaluate the RLP-regularizer in CIFAR-100 vs ciFAIR100 in Figure A.11. While the random label accuracy and training accuracy remain similar across the two datasets, the reduction in memorization induced by the RLP-regularizer is less detrimental on the train accuracy on ciFAIR-100 than on CIFAR-100. This indicates that the degraded generalization performance observed on CIFAR-100 is, at least in part, driven by train-test duplicates. Overall, this experiment further demonstrates that the proposed RLP metric and regularizer are effective tools for analyzing memorization phenomena.

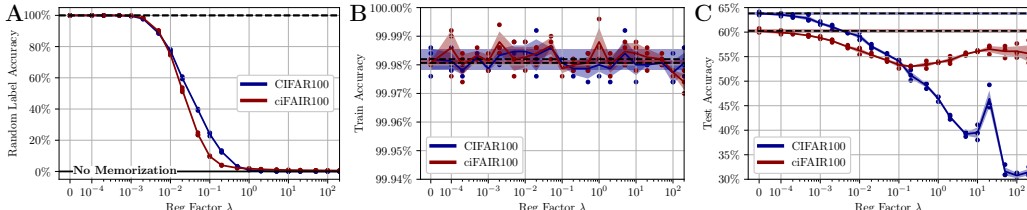

Figure A.11: WideResNet-16-4 on CIFAR-100 and ciFAIR100 (Barz & Denzler, 2020). Random label (**A**) train (**B**) and test (**C**) accuracy under RLP-regularization for different regularization factors λ.

## A.12 FELDMAN SCORES

We compare the random label accuracy as a measure of memorization to the memorization score proposed by Feldman & Zhang (2020).

Feldman & Zhang (2020) define memorization per sample by testing whether a model needs to be trained on a specific sample in order to correctly classify it. Concretely, they consider the change in a model's accuracy on each training dataset when the sample is either included in or excluded from the training set. Since an exact evaluation of this metric necessitates a full training run for each sample, the authors approximate it by removing 30 % of the training data at once and averaging results over multiple subsampled training runs to obtain a per-sample score. Even with this approximation, hundreds to thousands of full training runs are required. The resulting scores for ResNet-50 trained on CIFAR-100 are publicly available.

To compare against this method, we compute the random-label prediction accuracy per sample by averaging over 50 independently initialized training runs. Figure A.12 shows the distributions of the original Feldman scores, our reimplementation of their method, and random-label accuracy on CIFAR-100. The distributions differ clearly: the Feldman scores are bimodal, whereas the random-label accuracy is approximately Gaussian. Moreover, we find low correlation between the two measures (Pearson's $r = 0.08$ with $p < 10^{-8}$).

Despite this, we validate that the random label accuracy is highly sample-specific. We performed an Anderson–Darling test to reject the hypothesis that all samples share the same underlying distribution, and repeated independent estimates of random-label accuracy per sample exhibit high correlation (Pearson's $r \approx 0.9$). This confirms that random-label accuracy is indeed a stable property of each sample.

Although the lack of correlation between the two measures is counterintuitive, we argue that it is consistent with the previously observed lack of correlation between memorization and generalization for CIFAR-100.

Rather than directly measuring memorization, the Feldman score effectively measures whether the model can correctly classify a sample without having seen it, i.e., whether the model can generalize from the rest of the training set to that sample. It can be interpreted as constructing a one-sample validation set and comparing the model's performance on this sample to its performance on the training set. In this sense, the Feldman score directly measures generalization. It can also be viewed as quantifying the uniqueness of a sample within the dataset. For example, two very similar samples of the same class that are distinct from the rest of the dataset (similar to duplicates found between test and train sets by Barz & Denzler (2020) discussed in Appendix A.11) will not receive a high Feldman score even if these samples are memorized.

In contrast, generalization does not affect the random label accuracy due to the non-existent correlation between label and sample, thus providing a measure of memorization independent of a possible link between memorization and generalization. The random label accuracy measures whether memorization occurs, irrespective of whether that memorization is beneficial.

Thus, the initially surprising lack of correlation between random label accuracy and the Feldman scores in fact supports the hypothesis that reduced memorization and improved generalization are not directly coupled.

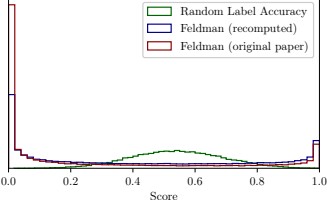

Figure A.12: Distribution of memorization scores proposed by Feldman & Zhang (2020) and distribution of per-sample random label accuracy on CIFAR-100. The original data from Feldman & Zhang (2020) were computed for ResNet-50, whereas our recomputed scores and random-label accuracy were derived using WideResNet-16-4.

## A.13 NOISY LABELS

To further analyze the effects of RLP-regularization on memorization, we examine the training accuracy on randomly labeled samples within the training dataset. These datapoints can only be predicted correctly through memorization. As shown in Figure A.13, the RLP-regularizer reduces the training accuracy on samples with noisy labels, while the accuracy on samples with intact labels remains high even for large regularization strengths. The proposed regularizer thus targets memorized samples specifically. This is particularly true for $\lambda = 10^5$, where at the same time the test accuracy simultaneously reaches its maximum. By limiting the memorization of incorrectly labeled examples, the generalization gap is reduced. This experiment highlights the strong connection between the random label accuracy of the RLP-head and training performance on noisy labels, as well as the effectiveness of the RLP-regularizer in mitigating memorization of noisy labels.

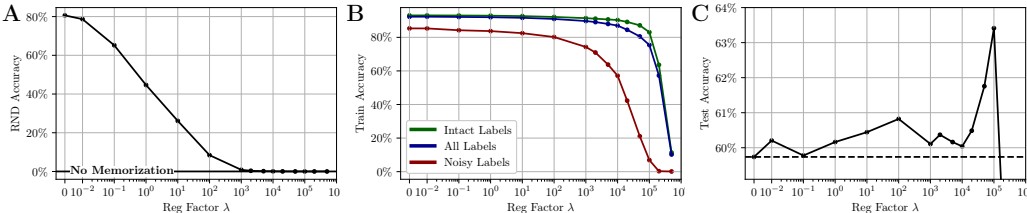

Figure A.13: ViT-B/32 on ImageNet with 10 % label noise. Random label accuracy (**A**), train accuracy (**B**), and test accuracy (**C**) for increasing regularization factors $\lambda$.

## A.14 MEMORIZATION UNDER OTHER REGULARIZERS

We compare the effectiveness of the random label accuracy as a measure of memorization against the more direct approach of measuring memorization via training accuracy on noisy labels. We conduct this comparison under several common regularizers: label smoothing (Figure A.14), dropout (Figure A.15), and weight decay (Figure A.16). Consistent with our observations in section 5.2, all regularizers lead to reduced random label accuracy, supporting the validity of this metric as an indicator of network complexity. At the same time, the random label accuracy proves to be a reliable measure of memorization when compared with the training accuracy on noisy-labeled datapoints. Our proposed memorization measure is fully non-intrusive and can be applied without altering the training data, unlike noisy label injection.

Comparing these results with those obtained when the RLP-regularizer is applied (shown for the same training setup in Figure A.13), we observe similar improvements in test performance. However, the primary purpose of the RLP-regularizer is not to specifically improve generalization, but to explicitly control memorization in order to study and better understand its underlying mechanisms and identify when and where reducing memorization leads to improved generalization. While other regularizers also reduce memorization effectively (e.g., weight decay, which drives both random label accuracy and noisy label training accuracy close to 0%, as seen in Figure A.16), the RLP-regularizer allows targeted application to arbitrary layers. This makes it particularly well-suited for studying the evolution of memorization within the network, as e.g. explored in section 5.5.

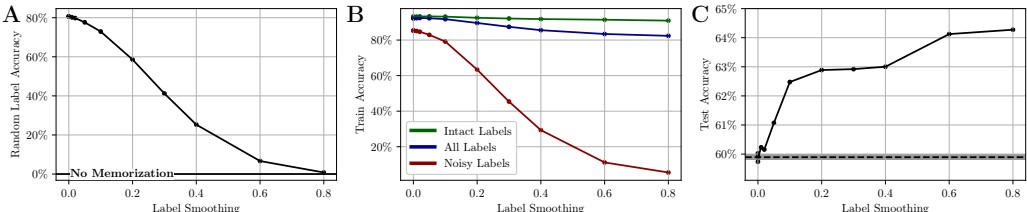

Figure A.14: ViT-B/32 on ImageNet with 10 % label noise and varying label smoothing strength.

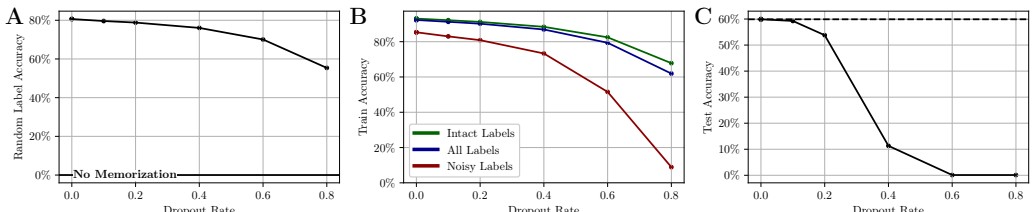

Figure A.15: ViT-B/32 on ImageNet with 10 % label noise and varying dropout strength.

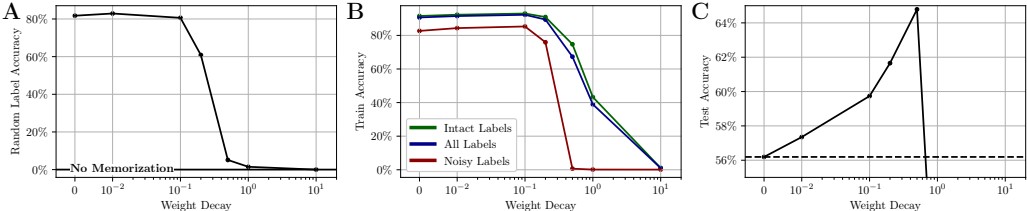

Figure A.16: ViT-B/32 on ImageNet with 10 % label noise and varying weight decay strength.

## A.15 FULL NETWORK RANDOM TRAINING

To further support the connection between the random label accuracy as an empirical memorization measure and Rademacher complexity, we compare the random label accuracy of the proposed RLP-head with the training accuracy achieved when training an entire network on random labels across varying network widths.

The training accuracy obtained under fully random labels closely resembles the Rademacher complexity. The only approximations involved include using SGD to obtain an approximately optimal model instead of taking the supremum over all models, extending the binary-label definition to a multi-class accuracy setting, and estimating the expectation over random labelings via a finite number of independent training runs.

When varying the width of a WideResNet-16-$w$, we observe a strong correlation between the performance on random labels when the full network is trained end-to-end on these labels and the random label accuracy measured using the RLP-head only while the main network is trained on correctly labeled data as done in the rest of this manuscript.

This comparison also demonstrates that the random-label accuracy of the RLP-head is primarily determined by the capacity of the feature extraction network, rather than by the capacity of the head itself as long as the RLP-head is chosen to be sufficiently large.

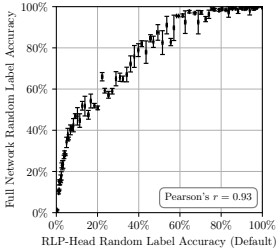

Figure A.17: WideResNet-16-$w$ on CIFAR-100, $n = 10,000$. Training the RLP-head only on random labels while the rest of the network is trained on class labels as performed in the rest of this manuscript compared against training the full network on random labels. Each datapoint represents a varying width factor $w$ similar to Figure A.9.

## A.16 MEMORIZATION WITH MIXUP

We evaluate the effect of mixup (Zhang et al., 2018b) on the random label accuracy. To do so, we apply mixup to the input images, class labels, and random labels during training, and then perform an additional epoch on the training set without mixup and without updating the weights in order to measure both the training accuracy and the random label accuracy. Since mixup is intended to reduce memorization, increasing the mixup strength (i.e., larger $\alpha$) indeed lowers the random-label accuracy. However, we still observe substantial memorization even at $\alpha = 10$.

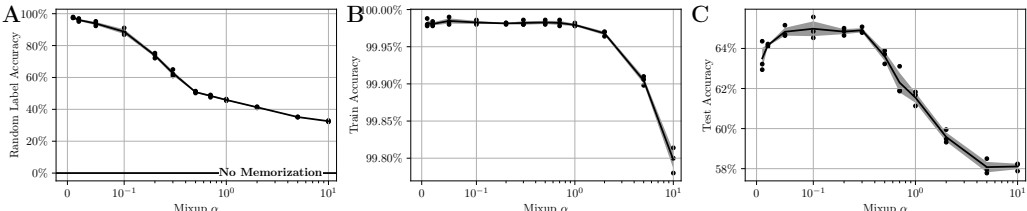

Figure A.18: WideResNet-16-4 trained on CIFAR-100 with varying mixup strength $\alpha$.

## A.17 ADVERSARIAL ROBUSTNESS

We evaluated the adversarial robustness of a ViT-B/32 trained on ImageNet with the RLP-regularizer under attacks by PGD (Madry et al., 2018), FGSM Goodfellow et al. (2018) and APGDT Croce & Hein (2020). We used default hyperparameters and $\sigma = 0.1$ for gaussian noise and $\epsilon = 1/255$ for the other attack methods. We report results for the difference of the accuracy under attack to the baseline (i.e., no RLP-regularizer; $\lambda = 0$) for various regularization factors $\lambda$ in Figure A.19 and the exact accuracies under attack for optimal $\lambda = 10^4$ in Table A.1. The RLP-regularizer improves the adversarial robustness in all scenarios.

Table A.1: Accuracy under adversarial attack of RLP-regularized models.

|  | Gaussian Noise | PGD | FGSM | APGDT |
|---|---|---|---|---|
| baseline ($\lambda = 0$) | $25.1_{\pm 0.3}$ | $16.6_{\pm 0.2}$ | $23.4_{\pm 0.3}$ | $17.6_{\pm 0.2}$ |
| RLP-regularized ($\lambda = 10^4$) | $26.2_{\pm 0.8}$ | $18.0_{\pm 0.4}$ | $24.9_{\pm 0.5}$ | $18.5_{\pm 0.4}$ |

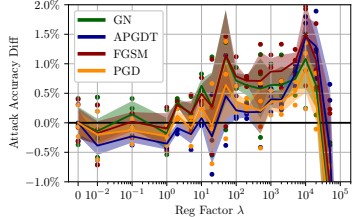

Figure A.19: Accuracy difference of RLP-regularized models to unregularized models under adversarial attacks.

## A.18 MEMBERSHIP INFERENCE ATTACKS

We perform membership inference attacks on our ViT-B/32 trained on ImageNet to assess if the RLP-regularizer leads to improved membership robustness. We use an MLP with 2 hidden layers (dimensions 512 and 256) to perform binary classification on the sorted logits of the models to determine if a sample was part of the training data or not. We train our attack model on the logits of the original model and create a balanced test dataset to evaluate the accuracy of the membership prediction. The attack model is trained for 10 epochs using the Adam optimizer with a learning rate of $10^4$ and a batch size of 256. We evaluate three independently initialized ViT models attacked by five independently initialized attack models each. While the model is not very vulnerable to membership inference attacks without RLP-regularization ($64.27\,\% \pm 0.05\,\%$), the RLP-regularizer further increases the robustness of the model reducing the membership accuracy to $62.27\,\% \pm 0.06\,\%$ for the regularization factor of optimal generalization $\lambda = 10^4$. Full results are reported in Figure A.20.

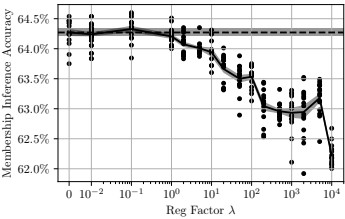

Figure A.20: Membership Inference Accuracy for ViT-B/32 models trained under RLP-regularization.

## A.19 MEMORIZATION OF LONG-TAIL SAMPLES

To test our hypothesis that memorization is beneficial for undersampled data distributions but not for sufficiently sampled ones, we conduct an additional experiment using a ViT-B/32 model trained on the ImageNet-LT (long-tail) dataset Liu et al. (2019b). ImageNet-LT is a subset of the original ImageNet dataset in which certain classes are deliberately undersampled in the training set, while the test set remains unchanged.
We compare test performance as a function of the number of training samples per class, evaluating models trained with and without RLP-regularization in Figure A.21. Under RLP-regularization, where memorization is stopped, lower test performance is reached for classes with low sample counts, i.e. classes with fewer than 80 training samples are not learned. For classes with higher sample counts, performance sometimes improves and sometimes does not.
These results support our hypothesis that memorization is useful in undersampled regimes and may or may not be in oversampled regimes.

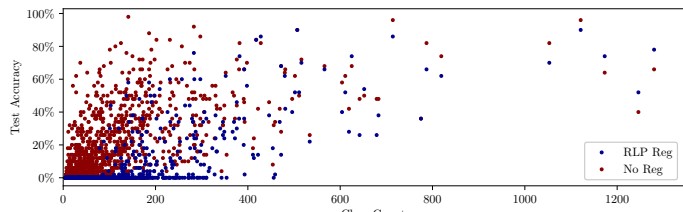

Figure A.21: ViT-B/32 trained for 300 epochs on ImageNet-LT without RLP-regularizer and with regularizer ($\lambda = 10^5$). Average test accuracy per class count.

## A.20 RLP-HEADS INSIDE TRANSFORMER LAYERS

To test our hypothesis that memorization is shifted into the transformer blocks under RLP-regularization applied to the output of all transformer blocks, we insert additional RLP-heads inside the transformer blocks: after the attention mechanism, after the first fully connected layer, and after the second fully connected layer of each transformer block. The corresponding results are shown in Figure A.22.

Although all additional heads detect memorization within the network, the largest values still appear at the output of each transformer block as measured in the rest of this manuscript. When applying RLP-regularization only to the final block head, memorization is reduced in the components of later transformer blocks, as illustrated in Figure A.22A.

However, when regularizing based on all RLP-heads, as in Figure 8B, we observe non-zero memorization in the last four transformer blocks after the attention mechanism and after the first fully connected layer (see Figure A.22B). Additional memorization may also be encoded within the representations inside the attention mechanism itself.

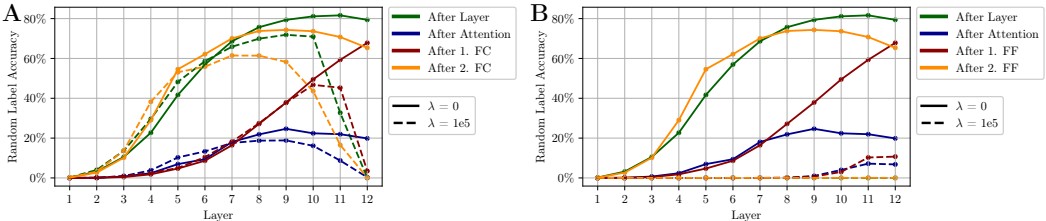

Figure A.22: Random label accuracy inside transformer blocks. **A**: Regularization based on the RLP-head attached after block 12 only. **B**: Regularization based on the RLP-heads attached after all blocks.

## A.21 FROZEN FEATURE EXTRACTOR AFTER REGULARIZATION

To verify that memorization is genuinely mitigated by the RLP regularizer—and not merely concealed from the specific RLP-head trained alongside it, we conducted an additional experiment using our ViT-B/32 setup on ImageNet. We train a new RLP-head as before, but keep the feature extractor frozen using the weights obtained from a previous training run with RLP-regularization (in the default setup). This setup tests whether a newly trained RLP-head can recover the random labels using the representations learned under regularization. Importantly, the random-label mapping between samples remains unchanged across the two runs. With a regularization strength of $\lambda = 10^{-4}$ applied during training of the feature extractor, the newly trained RLP-head achieves a random label accuracy of only $14.65\,\%$ (the baseline is approx. $70\,\%$, compare Figure 2A). This provides further evidence that the RLP-regularizer effectively mitigates memorization rather than merely obscuring it.

## A.22 TEXT CLASSIFICATION

We use the RLP-heads to study memorization in language tasks. Our experiments are conducted on Yahoo! Answers (Zhang & LeCun, 2016) using RoBERTa-base (Liu et al., 2019a) trained with Adam and a learning rate of $2 \cdot 10^5$ with cosine scheduling for 50 epochs. Results are shown in Figure A.23.

Interestingly, without regularization we observe pronounced memorization in the early layers, peaking at layer 7 and decreasing in later layers. This suggests that the model may be oversized for the given dataset.

While RLP-regularization applied via the RLP-head on the final (12th) transformer layer reduces memorization across all layers, we do not observe a shift of memorization towards later layers unlike the ViT results on ImageNet shown in Figure 8.

Furthermore, test accuracy does not improve with RLP-regularization, indicating that memorization may actually benefit generalization for the studied model and dataset.

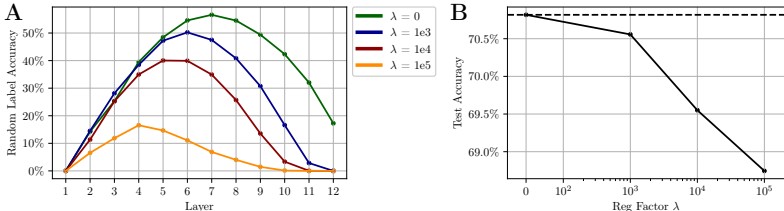

Figure A.23: RoBERTa-base trained on Yahoo! Answers with RLP-regularization based on the RLP-head attached to the final (12th) transformer layer only. **A**: Random label accuracy of RLP-heads at different layers. Memorization occurs in early layers and is reduced in later layers. **B**: The test accuracy does not improve when memorization is suppressed.

