# OpenReview forum: "Random Label Prediction Heads for Studying Memorization in Deep Neural Networks"
_ICLR.cc/2026/Conference — ICLR 2026 Poster_

### Official Review · Reviewer_Lg7F · 2025-10-29

**Soundness:** 3
**Presentation:** 2
**Contribution:** 3
**Rating:** 6
**Confidence:** 4

**Summary:**

This work proposes a new mechanism that both measures and regularizes memorization in ML models. The proposes mechanism introduces a loss term that measures how well a learned representation can be used to predict randomly assigned labels. Smaller loss is correlated with stronger memorization in the representation. The loss then can become a regularizer that prevents memorization. Empirical study shows that the  reduced memorization may have different effects on model utility depending on the sample size and the nature of information being memorized.

**Strengths:**

This paper proposes an interesting and effective mechanism based on simple principles to detect and control memorization. The design is interesting and the intuition is clear. Well done. The work is technically sound.

**Weaknesses:**

Despite the smart design of RLP-head and the fairly comprehensive experiments, there seem to be a few missing links in the argument of the paper. Specifically:

1) The paper proposes the loss on random label prediction as **both** the regularizer **and** the measure of memorization. Notice that the RLP loss serves as a proxy of empirical Rademacher complexity, which measures the model's capacity to memorize instead of the amount of memorization. Does a low RLP score necessarily mean less memorization?

2) Given the plethora of work on the pro and cons of memorization, I wonder if the empirical evaluation results and their implications are different from previous work. (See questions.)

In addition, the pdf file seems not fully follow the template. Some margins between the paragraphs are too small. Hope this can be fixed in future versions.

**Questions:**

1) Could you use a different metric for memorization, say influence-based heuristic in [1] or a method of your choice, to show that the model has less memorization when regularized more heavily with RLP?

2) Has empirical Rademacher complexity ever been used as regularizer before?

3) There has been literatures showing memorization could be beneficial (long-tail) [2] and detrimental (wrong-label) [3]. What are the key insights in this work's experiment that are different from the previous work?

[1] Feldman, Vitaly, and Chiyuan Zhang. "What neural networks memorize and why: Discovering the long tail via influence estimation." Advances in Neural Information Processing Systems 33 (2020): 2881-2891.

[2] Feldman, Vitaly. "Does learning require memorization? a short tale about a long tail." Proceedings of the 52nd annual ACM SIGACT symposium on theory of computing. 2020.

[3] Liu, Sheng, et al. "Early-learning regularization prevents memorization of noisy labels." Advances in neural information processing systems 33 (2020): 20331-20342.

---

> ### Author Response · Authors · 2025-11-21
>
> Thank you for your positive and constructive feedback.
>
> Weaknesses
> 1. Thank you for pointing out that the capacity to memorize is not the same as the memorization that actually occurs. To assess only the potential for memorization, training a full network on random labels as done, for example, by Zhang et al. (2021) would be appropriate. Motivated by exactly this challenge – measuring the memorization that actually occurs when training on real class labels – we introduced RLP-heads, which can be attached non-invasively during standard training.\
> To verify that our metric truly captures memorization, we conducted new experiments using a training dataset with 10% label noise. We separately evaluated the training performance on noisy and intact labels while varying the strength of the RLP-regularizer (Appendix A.13) as well as other regularizers (Appendix A.14). In all cases, we observed a strong correlation between the RLP-head’s random label accuracy and the model’s performance on the noisy- labeled datapoints. Since correctly predicting noisy labels necessarily requires memorization, these findings validate that our proposed metric indeed measures memorization.
> Additionally, in the newly added Appendix A.21, we show that for models in which memorization has been suppressed by the RLP-regularizer, a newly trained RLP-head is unable to recover the random labels when trained on the model’s frozen feature extractor. This demonstrates that the reduction in memorization persists even when the RLP-head used as a metric is not used for parallel regularization, providing further evidence that the memorization is genuinely reduced rather than merely hidden by the parallel application of the RLP-regularizer.
> 2. Please see question 3.
>
> Questions
> 1. We used the training performance on randomly labeled datapoints to demonstrate that the RLP-regularizer effectively reduces memorization, as shown in Appendix A.13.\
>  We also compare our memorization metric in depth to the metric proposed by Feldman et al. While their work presents interesting insights, we detail the substantial conceptual differences between their metric and ours. Rather than directly measuring memorization, the Feldman score effectively measures whether the model can correctly classify a sample without having seen it, i.e., whether the model can generalize from the rest of the training set to that sample. It can be interpreted as constructing a one-sample validation set and comparing the model’s performance on this sample to its performance on the training set. Thus, Feldman et al. do not explicitly determine whether memorization occurs, but rather whether it is beneficial. In this sense, the Feldman score differs from our metric, especially in scenarios where memorization does not directly harm generalization. Please refer to Appendix A.12 for a detailed discussion, including an empirical comparison between our proposed metric and that of Feldman et al.
> 2. To the best of our knowledge, empirical Rademacher complexity has not previously been used as a regularizer. However, we want to emphasize that the proposed RLP-regularizer is not an exact implementation of Rademacher complexity (which is not feasible to compute in real-world scenarios).
> 3. We agree with the work you referenced. In fact, we observe similar behavior on long-tail classes (see the added Appendix A.19). We added the work of Liu et al. to our related work. Our contribution lies in showing that RLP-heads provide a powerful yet simple tool for studying memorization. RLP-heads are lightweight and can be applied intervention-free in parallel to standard training procedures. Furthermore, their placement at specific layers enables layer-wise investigation of memorization, revealing how memorization evolves throughout the network and how it can be influenced by regularization.

---

### Official Review · Reviewer_hHXA · 2025-10-29

**Soundness:** 2
**Presentation:** 2
**Contribution:** 2
**Rating:** 2
**Confidence:** 5

**Summary:**

The paper proposes a method to measure memorization in deep models. The method involves a trainable RLP heads that can be attached to any layer of the network. The performance of RLP heads serves as a proxy for sample-level memorization. The authors also propose an RLP-based regularizer that reduces memorization by penalizing confident random-label predictions.

**Strengths:**

I think this is a really important problem, especially given the fact that sample-level memorization is expensive. The authors present a light weight solution, that can not only be run with relatively low latency, but can also be adapted to different layers of the model. I think this paper has a lot of potential if the authors can address the concerns below.

**Weaknesses:**

1. Lack of Suitable Baseline:

RLP approximates sample-level memorization. However, the paper lacks a clear baseline to verify whether the points identified or regularized by RLP are indeed the truly memorized samples. This is a major concern. The authors should validate this by comparing RLP’s behavior against established memorization benchmarks. This can be using Feldman et al.’s methodology or by introducing random noise images or “canary” points into the dataset. This would help determine whether 1) RLP selectively targets memorized data or unintentionally affects well-learned samples 2) It will also help understand how RLP behaves when base points are learned vs memorized. At this point, it is hard to gauge how well this technique performs. However, a good baseline can alleviate those concerns.

2. Limited Comparison with Existing Regularization Methods:

Although the authors present preliminary results suggesting that RLP-based regularization reduces memorization, the study lacks *direct* comparisons to standard regularizers (e.g., dropout, weight decay, or label smoothing). Such comparisons are critical for understanding whether RLP provides a distinct benefit beyond existing techniques. Ideally, these evaluations should be performed along two axes: (a) classification accuracy on intentionally mislabeled or noisy points (refer to point 1) to measure memorization control and (b) test accuracy on clean data (to assess generalization). Without these baselines, it is difficult to fully gauge the contribution and novelty of RLP as a regularizer.

3. Narrow Experimental Scope:

The current experiments are restricted to vision models trained on image datasets. Extending the analysis to text classification models and even LLMs would significantly strengthen the work, demonstrating the generality of RLP as a tool for studying memorization across architectures and domains.

**Questions:**

Refer to points above

---

> ### Author Response · Authors · 2025-11-21
>
> Thank you for your constructive feedback. We added multiple new experiments to address your concerns.
>
> Weaknesses
> 1. In Appendix A.13, we add 10% noisy labels as canary points to the training dataset and evaluate the training performance separately on the noisy and intact labels while increasing the strength of the RLP-regularizer. We find that the RLP-regularizer specifically targets the noisy (and thus memorized) samples, while simultaneously improving generalization.\
> We also compare our memorization metric in depth to the metric proposed by Feldman et al. While their work presents interesting insights, we identify substantial conceptual differences between their metric and ours. Rather than directly measuring memorization, the Feldman score effectively measures whether the model can correctly classify a sample without having seen it, i.e., whether the model can generalize from the rest of the training set to that sample. It can be interpreted as constructing a one-sample validation set and comparing the model’s performance on this sample to its performance on the training set. Thus, Feldman et al. do not explicitly determine whether memorization occurs, but rather whether it is beneficial. In this sense, the Feldman score differs from our metric, especially in scenarios where memorization does not directly harm generalization. Please refer to Appendix A.12 for a detailed discussion, including an empirical comparison between our proposed metric and that of Feldman et al.\
> Additionally, we perform an experiment on the ImageNet LT dataset in the added Appendix A.19 were we find that reducing memorization via the RLP-regularizer leads to reduced performance on rare classes in the long tail of the data distribution.
> 2. We conducted the suggested experiments comparing dropout, weight decay, and label smoothing against the RLP-regularizer, as reported in Appendix A.14. We find that all regularization techniques reduce memorization, as measured by the random label accuracy of the RLP-head and by the training accuracy on injected noisy-labeled datapoints. These two measures align closely, further supporting the validity of random label accuracy as a memorization metric.\
> While some of the other regularizers yield slightly better generalization performance, we want to emphasize that the proposed RLP-regularizer is not intended to replace established regularization techniques designed to improve generalization. Instead, its primary purpose is to provide a tool for studying and controlling memorization, thereby enabling a deeper understanding of its underlying mechanisms. This includes examining the relationship between memorization and generalization, as well as enabling targeted, layer-specific memorization reduction. We have clarified this intention in the main manuscript.
> 3. We additionally conduct experiments on text classification, demonstrating that RLP-heads can be used to study memorization effects beyond vision tasks. Interestingly, for a RoBERTabase model trained on the Yahoo! Answers text classification dataset, we observe that memorization emerges in early layers and then diminishes in later layers. Please refer to Appendix A.22 for the full experimental results and a detailed discussion

---

### Official Review · Reviewer_xMP4 · 2025-10-30

**Soundness:** 3
**Presentation:** 3
**Contribution:** 3
**Rating:** 6
**Confidence:** 4

**Summary:**

The paper proposes Random Label Prediction heads (RLP-heads) as a simple mechanism to measure and control memorization in deep nets during standard supervised training. Each training sample is given an auxiliary random label. A small prediction head attached to an intermediate activation is trained to predict these random labels in parallel with the main task. The authors interpret the RLP accuracy as an empirical proxy for Rademacher complexity and use it to study how memorization evolves over time and across layers. They also introduce a regularizer that penalizes correct random-label predictions to suppress memorization while keeping the task head untouched. Formally, the losses are L_{\text{class}}=-\log p_y, L_{\text{rnd}}=-\log \hat p_{\hat y}, and the regularizer L_{\text{reg}}=\log(1-\hat p_{\hat y}) scaled by \lambda.

Empirically, the paper shows:
1.	RLP accuracy tracks capacity and overfitting dynamics, rising to about 70 percent when training ViT-B/32 on ImageNet as test and train accuracies begin to separate.
2.	Standard regularizers like dropout, weight decay, and label smoothing reduce RLP accuracy, supporting the complexity interpretation.
3.	Using L_{\text{reg}} can reduce overfitting and improve test accuracy on ImageNet with ViT (about +1.5 points), but on CIFAR-100 with WRN it reduces memorization without helping test accuracy.
4.	Layer-wise probes reveal memorization grows with depth and that regularizing the final layer shifts memorization earlier rather than eliminating it.
5.	Dataset size and noise matter. Suppressing memorization helps when data are well sampled, but can hurt on undersampled datasets. Adding label noise makes the regularizer beneficial, as predicted.

**Strengths:**

•	Originality: using online random-label heads to continuously estimate per-layer memorization without retraining on random labels is clever and practical
•	Quality: solid empirical study across architectures, datasets, head designs, and hyperparameters, including offline sanity checks and dataset subsampling
•	Clarity: method, losses, and training protocols are well specified, figures communicate dynamics and layer-wise trends
•	Significance: provides a low-overhead diagnostic for memorization and a tunable knob to reduce it, yielding nuanced insights on when memorization helps or hurts generalization; the layer-wise shift phenomenon is especially interesting

**Weaknesses:**

•	The mapping from RLP accuracy to Rademacher complexity is argued empirically; a theoretical bridge or formal bound would strengthen the claim beyond correlation
•	Improvements on ImageNet are relatively small and sensitive to \lambda; practical guidance on choosing \lambda beyond grid search is limited
•	Baseline comparisons are missing to targeted anti memorization methods such as mixup, manifold mixup, early stopping with sharpness awareness, or confidence based noise filtering; the paper mostly compares to generic capacity regularizers
•	The regularizer may alter features in ways that indirectly affect the main head; while gradients are restricted, feature extractor changes can still trade off task signal vs sample specificity; stronger checks isolating collateral effects would help
•	The undersampling hypothesis is compelling but remains somewhat post hoc; additional controlled long tail benchmarks or per class sampling analyses would solidify it
•	Compute overhead and wall clock costs from extra heads, multi head variants, and per layer probes are not quantified in detail; practicality at large scale is uncertain
•	Privacy claims are hinted at in motivation but not evaluated; no extraction or canary tests are provided

**Questions:**

•	Can the authors formalize the connection between RLP accuracy and empirical Rademacher complexity for multiclass settings, perhaps via margin based surrogates or a bound that depends on head capacity and n?
•	How sensitive are results to the random label assignment itself; do multiple seeds produce similar \lambda optima and layer profiles, and what is the variance?
•	Could you compare RLP regularization against mixup or manifold mixup at matched hyperparameter tuning budgets, and report both accuracy and calibration?
•	Do RLP metrics predict robustness or privacy leakage; for example, do higher RLP accuracies correlate with membership inference or canary extraction rates?
•	On long tail datasets like iNaturalist or ImageNet LT, does suppressing memorization reduce performance on rare classes; per tail analysis would test the hypothesis directly
•	For the observed shift of memorization to earlier layers, can you probe within blocks to rule out within block hiding; e.g., multiple taps inside the same transformer block
•	What is the runtime and memory overhead of single head vs multi head vs all layers, and how does this scale with n and hidden size
•	Does RLP regularization interact with self supervised pretraining; does it help fine tuning stability or hurt transfer

---

> ### Author Response · Authors · 2025-11-21
>
> Thank you very much for your very detailed feedback and many suggestions.
>
> Questions
> 1. We agree that a formal bound connecting the RLP-head’s random label accuracy to Rademacher complexity would support our work. However, it is not straightforward to provide a bound on the random label accuracy that does not already rely on assumptions about memorization occurring within the feature extractor and that does not only bound the memorization capacity of the RLP-head itself. If one assumes that the feature extractor maps samples to sample-specific features, then a bound for the RLP-head could be derived; however, such a bound would reflect only the RLP-head’s ability to memorize the samples and would not capture the overall memorization capacity of the model.\
> While we acknowledge that this issue is inherently tied to the ambiguity between the capacity of the RLP-head and that of the full model, we provide extensive empirical analyses to disentangle these two factors. Specifically, we vary the capacity of the head (Appendix A.6), the number of random labels (Appendix A.4), and the optimization budget (Figure 2B). These results collectively support the validity of the RLP-head’s accuracy as a proxy for network memorization. Motivated by your feedback, we have also added a new experiment to further empirically connect our findings to Rademacher complexity:\
> In Appendix A.15, we compare the random label accuracy of the proposed RLP-head with the training accuracy achieved when training an entire network on random labels across varying network widths. The training accuracy obtained under fully random labels provides a close empirical approximation of Rademacher complexity. The only involved approximations are using SGD to obtain an approximately optimal model instead of taking the supremum over all models, extending the binary-label definition to a multi-class accuracy setting, and estimating the expectation over random labeling via a finite number of independent training runs. We observe a strong correlation between the performance on random labels when the full network is trained end-to-end on random labels and the random label accuracy measured using only the RLP-head while the main network is trained on correctly labeled data, as done in the rest of the manuscript.
> 2. The RLP-heads are not highly sensitive to the specific random label assignments. To illustrate this, we added multiple randomly initialized repetitions per data point in the figures of the main paper, which show consistently low variance and a reproducible optimum for $\lambda$ in Figure 4C.
> 3. We added an experiment in Appendix A.16 evaluating the effect of mixup (Zhang et al., 2018b) on generalization and memorization as measured by the random label accuracy. The RLP-head confirms that mixup reduces memorization, however, it does not eliminate it entirely. Simultaneously, it improves generalization performance. We want to emphasize that the RLP-regularizer is not primarily intended as a replacement for established memorization-reduction or generalization-improving regularizers. Instead, it is meant as a tool for studying and understanding the effects of targeted, layer-specific memorization reduction.
> 4. As suggested, we conducted additional experiments demonstrating increased robustness against adversarial attacks and membership-inference attacks. Please refer to Appendix A.17 and Appendix A.18 in the revised manuscript for a detailed analysis.
> 5. In a per-tail analysis of the ImageNet LT dataset, we indeed find that reducing memorization via the RLP-regularizer leads to reduced performance on rare classes. Please see Appendix A.19 in the revised manuscript. Thank you for suggesting this experiment which provides valuable evidence supporting the hypothesis of our work.
> 6. A detailed analysis of RLP-heads attached to intermediate transformer layers reveals that memorization persists even when regularization is applied using RLP-heads after all transformer layers. This observation is reported in the new experiment described in Appendix A.20.

---

> ### Author Response · Authors · 2025-11-21
>
> 7. We summarize the additional compute and memory requirements for a ViT-B/32 trained on ImageNet with a batch size of 2 on a single NVIDIA 4090 GPU below:
> | Mode            | Time per iteration (ms) | VRAM (MiB) |
> |-----------------|-----------|--------------|
> | no RLP-head | 10.60     | 3173         |
> | RLP-head at last layer       | 11.50     | 5725         |
> | RLP-heads at all layers        | 47.90     | 24069        |
>
> While the overhead introduced by a single RLP-head at the final transformer layer is modest, attaching RLP-heads to all layers leads to a substantial increase in both compute time and memory usage - though it remains manageable in practice. The overhead scales linearly with the number of random labels $n$. We emphasize that our method is not intended to achieve new state-of-the-art performance results. Instead, it is designed as a tool for analyzing and studying memorization, for which we argue that the introduced overhead is acceptable.
> 8. While we are interested in exploring the effects of memorization during self-supervised training in future work, the RLP-head structure proposed here is specifically designed for supervised classification tasks.
>
> Thank you again for suggesting many insightful experiments which significantly strengthened our contribution!

---

### Official Review · Reviewer_BWFD · 2025-11-01

**Soundness:** 3
**Presentation:** 3
**Contribution:** 3
**Rating:** 6
**Confidence:** 4

**Summary:**

The paper introduces Random Label Prediction (RLP) heads: auxiliary prediction heads attached at various depths that are trained to predict fixed, per-example random labels during standard supervised training. The accuracy of these heads is used as a proxy for memorization/capacity. The authors also propose an RLP regularizer that discourages the feature extractor from confidently fitting the random labels, aiming to limit memorization without changing the task head. Experiments on ViT-B/32 with ImageNet and WRN-16-4 with CIFAR-100 show: (i) RLP accuracy rises early in training and can reach high levels on ImageNet; (ii) standard regularizers like dropout, weight decay, and label smoothing tend to reduce RLP accuracy; (iii) the RLP regularizer can improve ImageNet generalization but can hurt on CIFAR-100; (iv) RLP attached at intermediate layers suggests memorization increases with depth, and regularizing only the last layer can shift memorization earlier.

**Strengths:**

1. Simple, general mechanism that is easy to add and interpret: higher random-label accuracy indicates more sample-specific information in the features.

2. Clear empirical phenomena: early rise of RLP accuracy and monotonic increase with depth.

3. Useful bridge from measurement to control: correlation with common capacity controls and a targeted regularizer derived from the same signal.

4. Layerwise analysis reveals memorization shifting under last-layer regularization and provides a way to localize where class information emerges.

**Weaknesses:**

1. Theoretical grounding is informal. The connection to Rademacher complexity is motivational rather than a formal result; no theorem guarantees RLP accuracy is a calibrated surrogate for capacity in deep nets.

2. Attribution is ambiguous. The metric may conflate memorization in the feature extractor with the auxiliary head’s own capacity, the number of random labels, and optimization budget. Appendix ablations help but a clearer identifiability story would be better.

Minor comments
1. Please switch to the official ICLR template and fonts; the current submission uses a nonstandard font.

**Questions:**

1. CIFAR100 is known to have near duplicates in train and test sets, this could be causing the contrary results on CIFAR100, would de-duplicated CIFAR100 be a better fit?

---

> ### Author Response · Authors · 2025-11-21
>
> Thank you for your positive and constructive feedback.
>
> Weaknesses
> 1. Thank you for highlighting the need for greater clarity regarding the theoretical grounding. We confirm that the connection to Rademacher complexity is intended to be motivational and conceptual, providing the theoretical inspiration for using random label accuracy as a proxy for capacity and memorization. We did not intend to claim a formal guarantee linking the proposed RLP-head’s random label accuracy to Rademacher complexity. Our primary contribution lies in introducing the RLP-head metric and regularizer as novel empirical tools for studying memorization and demonstrating their utility through an extensive empirical study. We have clarified this distinction between theoretical motivation and empirical contribution more clearly in the revised manuscript.\
> To further empirically bridge our practical RLP-head metric and the Rademacher complexity, we included an additional experiment: In Appendix A.15, we compare the random label accuracy of the proposed RLP-head with the training accuracy achieved when training an entire network on random labels across varying network widths. The training accuracy obtained under fully random labels provides a close empirical approximation of Rademacher complexity: The only involved approximations are using SGD to obtain an approximately optimal model instead of taking the supremum over all models, extending the binary-label definition to a multi-class accuracy setting, and estimating the expectation over random labeling via a finite number of independent training runs. We observe a strong correlation between the performance when the full network is trained end-to-end on random labels and the random label accuracy measured using only the RLP-head while the main network is trained on correctly labeled data, as done in the rest of the manuscript.
> 2. We agree that the distinction between the RLP-head’s capacity and memorization is not completely unambiguous. However, we believe that our ablation experiments address all the aspects you mentioned - namely, the head’s own capacity (Appendix A.6), the number of random labels (Appendix A.4), and the optimization budget (Figure 2B) - providing strong empirical evidence that the observed effects are indeed driven by the network’s memorization. Unfortunately, we missed the correct font setting in the template. We corrected this in the updated version.
>
> Questions
> 1. As suggested, we additionally evaluated our regularizer on the de-duplicated variant of CIFAR-100, ciFAIR100, proposed by Barz & Denzler (2020), as described in Appendix A.11. While the random label accuracy and training accuracy remain similar across the two datasets, the reduction in memorization induced by the RLP-regularizer is less detrimental to the test accuracy on ciFAIR100 than on CIFAR-100. This suggests that the degraded generalization performance observed in CIFAR-100 is, at least in part, driven by train–test duplicates. Thank you for suggesting this insightful experiment!

---

### Author Response · Authors · 2025-11-21
**Rebuttal Summary**

We thank the reviewers for their detailed and constructive feedback, which has helped us substantially strengthen the manuscript.\
We have revised the main text to improve clarity and comprehensibility in response to their comments. In particular, we now state more explicitly that Rademacher complexity serves as a conceptual motivation for our method and we do not claim any theoretical guarantee linking the proposed RLP- head’s random label accuracy to Rademacher complexity. Nevertheless, we validate the conceptual similarities through a wide range of empirical investigations.\
We also clarify that the RLP-regularizer is intended primarily as a tool for studying and controlling memorization in order to better understand its underlying mechanisms, and not as a replacement for established regularization techniques aimed at improving generalization.


We revised the following figures in the main manuscript:
- Figure 4: We added multiple iterations per data point showing low variance of the random label and test accuracy.
- Figure 7C: We identified and corrected a minor bug in our label noise implementation for distributed training. After fixing it, the overall conclusion remains unchanged: the
regularizer provides substantial generalization improvements under noisy labels. We also added additional data points.
- Figure 9C: We added multiple iterations per data point.


Furthermore, to fully address the questions of the reviewers, we added the following sections to the appendix:
- A.11: Demonstrates that reducing memorization with the RLP-regularizer is less harmful when duplicate images between the train and test sets are removed using the ciFAIR100 dataset.
- A.12: Provides a detailed comparison between our method and the approach of Feldman & Zhang (2020).
- A.13: Shows strong similarities between our proposed random label memorization measure and the training accuracy on noisy labeled images, which act as canary points for memorization.
- A.14: Evaluates other regularizers (dropout, label smoothing, and weight decay) in terms of random label accuracy, performance on noisy-label canary points, and generalization, and compares them to the RLP-regularizer.
- A.15: Provides empirical evidence strengthening the link between Rademacher complexity and the RLP-head’s random-label accuracy by showing strong correlation with the accuracy of networks trained on fully random labels.
- A.16: Investigates memorization in models trained with mixup using random label accuracy.
- A.17: Demonstrates improved adversarial robustness in models trained with RLP-regularization.
- A.18: Demonstrates improved robustness against membership inference attacks in models trained with RLP-regularization.
- A.19: Shows that the memorization reduction induced by RLP-regularization, decreases performance particularly on long-tail classes.
- A.20: Validates our hypothesis that applying regularization to each transformer block’s output shifts memorization into the transformer blocks.
- A.21: Provides evidence for the memorization-mitigation effect of the RLP regularizer by showing that, for models in which memorization is suppressed by the regularizer, a newly initialized RLP-head is unable to recover the random labels when trained on the model’s frozen feature extractor.
- A.22: Demonstrates that RLP-heads can be used to study memorization effects beyond vision tasks by investigating memorization in text classification.

---

### Author Response · Authors · 2025-12-01

Dear AC,

We believe the reviewers' feedback has helped us substantially improve our manuscript.\
We have clarified the central focus of our work - an extensive empirical study of memorization based on our novel yet simple technique of RLP-heads. With the eleven new experiments added, including all those suggested by the reviewers, the manuscript has become a thorough and detailed empirical analysis of memorization and of RLP-heads as a tool to study it.\
All additional experiments requested by the reviewers support our hypotheses. Reviewers BWFD, xMP4, and Lg7F were already positive in their initial reviews, and the new experiments provide additional support for their assessments.\
The experiments on the de-duplicated CIFAR100 dataset suggested by BWFD align well with our findings.
Based on the suggestions of reviewer xMP4 we conducted six additional experiments that substantially strengthened the manuscript. Especially, the experiments on the ImageNet long-tail dataset strongly support our main observations.\
We also addressed the main concern of reviewer Lg7F regarding the connection between memorization and the RLP-head. In Appendix A.13, we show that applying the RLP-regularizer reduces memorization of noisily labeled (and thus memorized) samples, directly addressing this point.\
This issue was also a central concern of reviewer hHXA. While reviewer hHXA's initial review did not favor acceptance, the reviewer emphasized the potential of the manuscript if the raised concerns were addressed. We fully followed the reviewer's suggestions and were able to resolve all three of the raised issues.
The proposed experiments on noisily labeled datapoints (canary points) demonstrate that our RLP-regularizer effectively reduces memorization. We also provide a detailed comparison between our metric and the metric proposed by Feldman et al. Furthermore, we expanded the comparison with other regularization methods as suggested, highlighting the suitability of the RLP-regularizer for reducing memorization as well as the usefulness of the RLP-head's random label accuracy as a measure of complexity and memorization.
We broadened the scope of our work by adding experiments on text classification. These experiments reveal interesting patterns regarding the emergence of memorization in early layers and demonstrate that RLP-heads are also suitable for studying memorization in the NLP domain.

While we would have been curious about further feedback of the reviewers and eager to further improve our manuscript, the updated manuscript represents a substantial improvement over the initial submission.\
Our proposed simple yet effective tool for studying memorization in ANNs has allowed us to reveal numerous interesting findings and holds significant potential for future investigations into the effects and emergence of memorization.

Best regards,
The authors

---

### Meta-Review · Area_Chair_pSDv · 2026-01-13

**Summary:**

Three reviewers (BWFD, xMP4, Lg7F) are consistently positive and see the core contribution as a simple, broadly applicable empirical tool (RLP-heads) plus a targeted memorization regularizer with interesting layerwise phenomena. However, the submission was initially held back by concerns around:

- Theory/interpretation: The link to Rademacher complexity is motivational and not formal; reviewers worried the paper might overclaim or that RLP accuracy conflates head capacity/optimization with true memorization.

- Validation against memorization baselines: hHXA's main objection was lack of a strong baseline to verify that RLP targets memorized vs learned points (e.g., canaries/Feldman-style analyses).

- Comparative evaluation and scope: missing comparisons to other anti-memorization methods (mixup etc), limited non-vision evidence, and unclear practicality at scale (overhead).

Practical impact/robustness of conclusions: gains on ImageNet are relatively modest and sensitive to \lambda. CIFAR-100 results were counterintuitive and might be affected by duplicates/long-tail effects.

Overall, the contribution is primarily methodology + empirical characterization, not SOTA performance. With rebuttal evidence, the work is better supported, but it remains somewhat theory-light and the "what exactly is being measured" question is only empirically addressed.

**Reviewer Concerns:**

Concerns that were addressed well:

- Overclaiming about Rademacher complexity: Authors explicitly clarify it is conceptual motivation, not a theorem-level guarantee.

- Baseline validation of "memorization": Added noisy-label canary experiments (A.13) showing RLP-regularizer preferentially reduces performance on noisy labels (which require memorization), and comparison to Feldman & Zhang-style perspective (A.12). This directly targets hHXA/Lg7F's key worry.

- Comparison to standard regularizers: Added dropout / weight decay / label smoothing comparisons (A.14), relating them to both RLP accuracy and noisy-label canary behavior.

- CIFAR-100 duplicates hypothesis: Added ciFAIR100 results (A.11) supporting reviewer BWFD’s suspicion that duplicates partly explain the CIFAR-100 behavior.

- Variance / sensitivity to random labels: Added multi-run/low-variance evidence in main figures (per rebuttal summary).

- Scope expansion and “hypothesis stress-tests”: Added long-tail (ImageNet-LT) analysis (A.19), robustness to adversarial and membership inference attacks (A.17/A.18), layer/transformer-block probing (A.20), and text classification evidence (A.22).

- Compute overhead transparency: Provided concrete timing/VRAM numbers for last-layer vs all-layer heads.

Concerns that are still outstanding or only partially addressed:

Formal identifiability/theory: Even with clarifications and the correlation-to-random-label-training experiment (A.15), the central interpretation remains empirical; there is still no crisp statement of when RLP accuracy is guaranteed to reflect memorization in the backbone rather than probe capacity/optimization artifacts.

- Positioning vs existing memorization literature: Better now, but a reviewer could still argue the “key new insight” is mainly the tooling (layerwise, online, low overhead) rather than fundamentally new phenomena; the paper should be careful in how strongly it claims novelty of conclusions.

- Practical guidance / usability: \lambda sensitivity is acknowledged but still not fully resolved (beyond empirical sweeps); overhead for “all layers” is large, so recommended best practices could be sharper.

- Comparisons to targeted alternatives: Mixup is now included (A.16), but broader "matched budget" comparisons to other targeted anti-memorization approaches are still not exhaustive (though this is less critical if the paper frames itself as a measurement/control lens rather than a competing regularizer).

**Reviewer Scores:**

Reviewer BWFD would likely stay at 6 but might be nudged to 7 too. Their main asks were: clarify theory-as-motivation, address head-capacity ambiguity, fix template, and test de-duplicated CIFAR100. The rebuttal directly did these (especially ciFAIR100 + added empirical bridge).

Reviewer xMP4 might have moved from 6 to 7. Most of xMP4's concerns were about the missing pieces (mixup comparison, robustness/privacy linkage, long-tail per-class, within-block probing, overhead quantification, variance across seeds). The rebuttal adds most of these. The theory/guidance issues still remain however.

Reviewer hHXA might have moved from 2 to 4 (or maybe 5), but not necessarily to accept. hHXA's rejection hinged on lack of baseline validation, lack of comparisons to standard regularizers, and narrow scope. The rebuttal addresses each head-on (canaries + Feldman comparison + regularizer comparisons + text). Still, given their very high confidence and initial skepticism, they may remain unconvinced about identifiability/theory.

Reviewer Lg7F would likely stay at 6 or might have moved to 7. Their key conceptual worry was "capacity to memorize vs memorization that occurs", plus positioning vs prior work. The noisy-label/canary alignment and the "new head can’t recover labels on frozen features" experiment (A.21) are strong direct responses, making an upward nudge plausible.

Overall, a proper discussion would likely converge to 3 reviewers at 6-7, and one reviewer at 4 or 5.

Given this, I would recommend the paper for (weak) acceptance as it does contribute to solid empirical tool-building but it should also tone down its theoretical claims appropriately in the revision.

---

### Decision · Program_Chairs · 2026-01-26

Accept (Poster)